# From Ancient Fermentations to Modern Biotechnology: Historical Evolution, Microbial Mechanisms, and the Role of Natural and Commercial Starter Cultures in Shaping Organic and Sustainable Food Systems

**DOI:** 10.3390/foods14244240

**Published:** 2025-12-10

**Authors:** Yasmin Muhammed Refaie Muhammed, Fabio Minervini, Ivana Cavoski

**Affiliations:** 1Dipartimento di Scienze del Suolo, della Pianta e degli Alimenti, Università degli Studi di Bari Aldo Moro, 70126 Bari, Italy; yasmin.muhammed@uniba.it; 2Centre International de Hautes Etudes Agronomiques Méditerranéennes (CIHEAM)—Mediterranean Agronomic Institute of Bari, 70010 Valenzano, Italy; cavoski@iamb.it

**Keywords:** natural starters, commercial starters, fermentation, yeasts, lactic acid bacteria, organic food

## Abstract

From the first spontaneous fermentations of early civilizations to the precision of modern biotechnology, natural starter cultures have remained at the heart of fermented food and beverage production. Composed of complex microbial communities of lactic acid bacteria, yeasts, and filamentous fungi, these starters transform raw materials into products with distinctive sensory qualities, extended shelf life, and enhanced nutritional value. Their high microbial diversity underpins both their functional resilience and their cultural significance, yet also introduces variability and safety challenges. This review traces the historical development of natural starters, surveys their global applications across cereals, legumes, dairy, vegetables, beverages, seafood, and meats, and contrasts them with commercial starter cultures designed for consistency, scalability, and safety. Within the context of organic food production, natural starters offer opportunities to align fermentation with principles of sustainability, biodiversity conservation, and minimal processing, but regulatory frameworks—currently focused largely on yeasts—pose both challenges and opportunities for broader certification. Emerging innovations, including omics-driven strain selection, synthetic biology, valorization of agro-industrial byproducts, and automation, offer new pathways to improve safety, stability, and functionality without eroding the authenticity of natural starter cultures. By bridging traditional artisanal knowledge with advanced science and sustainable practices, natural starters can play a pivotal role in shaping the next generation of organic and eco-conscious fermented products.

## 1. Introduction

In recent times, there has been a noticeable change in consumer behavior towards organic food consumption, primarily fueled by a growing consciousness surrounding sustainable and healthy dietary choices. The global market for organic produce is valued at 90 billion euros [1]. In 2022, Italy had over 2.3 million hectares of land destined for organic agriculture and a total of over 30,000 organic operators, as reported by the regional Biobank Open Project and the national information system for organic agriculture (Sistema d’Informazione Nazionale sull’Agricoltura Biologica). This resulted in an annual domestic market value for organic products of 3.55 billion euros. Consumers perceive food from organic agriculture as natural and minimally processed [2,3]. Organic food products usually contain fewer pesticide residues and lower levels of heavy metals when compared to food from conventional agriculture [4,5,6,7].

Amidst this burgeoning trend, fermentation has emerged as a prominent method of careful food processing epitomizing the fundamental rules of EU about organic food (Reg. EU 848/2018) and harnessing the power of natural starters to allow food transformation. Food fermentation, with a history dating back to at least 6000 years, offers remarkable sensory attributes [8,9] and, if well controlled, does not need additives to ensure its safety [10,11] and long shelf life with low risk of spoilage [12,13,14]. Fermentation exploits the enzymatic activities of beneficial microorganisms (named “starters”) to convert organic substrates in simpler compounds, such as ethanol and organic acids. These starters include yeasts for alcoholic fermentation, *Bacillus* spp. for alkaline-fermented foods, and lactic acid bacteria (LAB) for fermentations yielding lactic acid as the main compound. Fermentation enabled our ancestors, across diverse regions, to overcome challenging seasons and environmental constraints by enhancing the preservability and, although not automatically, safety of food. Today, fermented foods and beverages encompass a diverse array of products enjoyed worldwide, including alcoholic beverages, dairy products, pickled vegetables, sauerkraut, cured sausages, and soy-based products [9,15]. Approximately one-third of the food produced globally is fermented [16].

Fermentation fits well with the EU regulations on organic food production because it often allows for the avoidance of the use of chemically synthesized additives, which are prohibited by those regulations. Embracing fermentation as a core processing technique in the organic food industry not only provides consumers with an extensive range of organic food choices endowed with appreciable sensory traits, but also aligns perfectly with the EU’s vision of promoting a holistic and environmentally conscious approach to food production and consumption. Traditional fermentation processes remain relevant as alternatives in those areas where modern preservation methods, like refrigeration, are absent [17,18,19,20,21]. However, fermentation, differently from refrigeration, unavoidably changes composition, flavor, and texture, of food, and cannot be applied to matrices rich in lipids. Indeed, lipids may be oxidized by microorganisms to aldehydes, ketones, and peroxides, producing off-flavors, rancid odors, and texture changes [22].

Previous review papers compared natural starters, consisting of preparations that harbor mixed and mostly undefined populations of pro-technological microorganisms, with commercial starters, consisting of selected and standardized microbial strains [23]. However, they focused just on one [24] or two [25] types of fermented food. Corbo et al. [26] examined several fermented food items, questioning the opportunities to use microorganisms isolated from natural starters to replace commercial starters. However, they did not primarily focus on comparison between natural and commercial starters and considered the fermented food items, regardless of the origin (whether from conventional or organic farming) of the raw matter/food ingredients.

The main objective of the current review is to provide a comprehensive and integrative overview of the evolution, microbial ecology, and technological applications of natural and commercial starter cultures in food fermentation, with particular emphasis on their role in organic food production systems. Specifically, the review aims to (i) trace the historical development of fermented foods and starter cultures, (ii) describe the microbial mechanisms governing natural starters with sourdough as a model, (iii) compare the functionality and safety of natural versus commercial starters, (iv) examine regulatory frameworks and organic certification requirements, and (v) explore future directions for harnessing natural microbial diversity in sustainable food processing.

### Methodology of Literature Search

This review was conducted, following a structured narrative approach, to summarize and critically evaluate current knowledge on natural and commercial starter cultures in organic food fermentation. Scientific articles were retrieved from major databases, including Scopus, Web of Science, PubMed, and Google Scholar, using combinations of the following keywords: “fermented foods”, “starter cultures”, “natural starters”, “commercial starters”, “organic fermentation”, and “regulation.” Searches were limited to publications in English, from 2000 to 2025, supplemented with earlier seminal works for historical context. Additional relevant documents were identified through the reference lists of selected papers and official regulatory sources (e.g., European Food Safety Authority, EU Commission and U.S. Department of Agriculture). Studies were included based on their relevance to the microbiology, technological aspects, and regulatory context of fermentation.

## 2. Fermented Food Items Through the Ages: The Cradles of Natural Starters

The history of natural starters can be traced back to the dawn of human civilization when the discovery and utilization of fermentation as a method of food preservation became an essential aspect of early practices of food processing [22,27,28]. Certain authors have proposed that fermentation techniques and their resulting products were likely developed approximately 10,000 years ago with the goal being to preserve food during scarcity while enhancing its taste [29,30]. Natural starters have played a pivotal role in traditional fermentation techniques across diverse cultures, boasting a remarkable history spanning millennia. During ancient times, fermentation was initiated by the action of microorganisms naturally present in the environment [31]. These complex microbial communities, consisting of bacteria, yeasts, and filamentous fungi (i.e., molds) conducted the fermentation, ultimately contributing to the unique flavors, aromas, and textures of the final products [32]. Early civilizations unknowingly noticed the power of microorganisms, allowing for the spontaneous fermentation of foods and beverages. As human knowledge advanced, the intentional selection and cultivation of specific starter cultures emerged, leading to the deliberate use of well-adapted microorganisms in fermentation [33,34]. This deliberate selection and use of natural starters, influenced by indigenous knowledge and cultural practices, has shaped the development and refinement of fermented foods and beverages throughout history.

### 2.1. Fermented Cereals and Legumes

The fermentation of cereals and legumes has left an indelible mark on many cultures. Microorganisms that naturally contaminate these matrices are able to convert them into a diverse range of staple, delicious, and healthy food products, such as bread and porridge [35]. The art of fermentation has also enriched the global culinary landscape with its diverse range of textures and flavors [36,37]. Examples of traditional fermented foods prepared from cereals and legumes in different parts of the world are listed in Table 1.

#### 2.1.1. Sourdough Bread

The production of sourdough bread has been performed since ancient times [52]. The microorganisms that drive fermentation are harbored in starters known as “levain” or “mother doughs”. Archeologists unearthed the oldest known leavened bread, which dates back more than 5000 years, during an excavation in Switzerland [53]. However, it was mentioned that it could be dated as far back as 10,000 years BC [52,54]. Egypt is believed to be the birthplace of sourdough bread, as people living there noticed an increase in volume when they left the mixture of water and wheat flour for some time and then, they deliberately added this mixture to a new dough, observing that the fermentation was thus enhanced. From Egypt, knowledge about sourdough bread was gradually spread throughout Greece and the Roman Empire. Egyptians also utilized beer foam in bread making [55], as beer foam contains metabolically active microorganisms—primarily yeasts—that initiate a new fermentation when added to dough.

In Greece, bread initially served as a food for home consumption in affluent households, with women handling its preparation. Later, evidence suggests the emergence of bakers, possibly organized in guilds, who produced bread for retail sale. The Greeks made remarkable improvements to technology and baking equipment [56]. The adoption of sourdough in Greece is believed to have occurred around 800 BC [55]. Greek gastronomy boasted a rich variety of bread types, including both sweet and savory types, made with different cereals and prepared using various techniques. Bread held significant dietary importance during the Roman Empire, leading to the establishment of numerous public bakeries. Bakers themselves became public officials, employed by the State to produce substantial quantities of bread that were distributed freely among Roman citizens. The profession of being a baker became a family tradition, passed down from one generation to the next. Greeks were known to offer votive offerings consisting of flour, cereal grains, or toasted bread and cakes mixed with oil and wine. During rituals dedicated to Dionysus, the god of fertility, joy, and passion, large loaves of bread were presented by priestesses. The transition from sacrificial to curative bread occurred swiftly, as patients visiting temples dedicated to Asclepius would leave bread imbued with the healing power attributed to the god and would receive a portion of it upon leaving [53]. The history and social significance of sourdough find rich documentation in countries such as France, Italy, and Germany, where this traditional biotechnology remains widely practiced.

#### 2.1.2. Miso and Natto

Miso and natto are original to Asia. Miso is a fermented soybean paste, whose roots trace back over a thousand years in Chinese cuisine. The earliest form of miso, known as “chiang” was initially made with fish, shellfish, and game. It dates back to before the Chau dynasty (722–481 BC). By the fourth century AD, chiang evolved into its soybean-based variant, miso, which became more common than chiang. Miso was then introduced to Japan between 540 and 552 AD, where it became a staple fermented food [46]. The Japanese adopted and adapted these techniques to suit their own tastes and ingredients. In Japan, it is called miso; in China, it remains known as chiang. In other Asian countries it was named differently: tauco in Indonesia, jang or doenjang in Korea, taochieo in Thailand, and tao-si in the Philippines [57].

Initially, miso was prepared by mashing soybeans and fermenting them with salt and other ingredients in wooden barrels or pots [58]. This early form of miso was known as “naan” or “isobe miso.” However, it was the introduction of koji, a specific natural starter dominated by the filamentous fungus *Aspergillus oryzae*, that revolutionized the miso-making process. Koji mold had been used in China for centuries to ferment various foods [59], including soy sauce and sake. In Japan, the use of koji in miso production is attributed to the Zen Buddhist priest Kakushin, who is said to have brought the knowledge of koji from China in the 14th century [60]. The incorporation of koji into miso making allowed for more controlled and efficient fermentation. This innovation marked the beginning of what is now recognized as traditional Japanese miso. Over the centuries, miso has continued to evolve, with regional variations and diverse ingredients. The production of miso now primarily involves soybeans, rice, salt, and koji mold. Today, there are numerous types of miso, each with its unique flavor and character, depending on factors such as the type of koji mold used, the proportions of soybeans to rice, and the duration of fermentation. Miso remains a cornerstone of Japanese cuisine, finding its usage into soups, marinades, sauces, and countless other dishes [47].

Natto, a distinctive Japanese fermented soybean dish, has a long history spanning centuries. It is believed to have originated in Japan around 2000 BC [61]. However, its exact origins are unclear, with various theories and historical accounts suggesting different developments. One theory posits that natto was accidentally discovered by Japanese villagers who left cooked soybeans in warm, humid conditions that favored fermentation by naturally contaminating microorganisms. Another theory suggests that Buddhist monks from China’s Yunnan province introduced natto to Japan during the Nara period (710–794 AD) [62,63]. The most widely accepted theory is that natto was discovered accidentally around the 11th century during the Heian period (794–1185 AD). The fermentation of natto relies on the bacterium *Bacillus subtilis*, which gives natto its characteristic slimy texture and strong aroma. Traditionally, natto was made by wrapping cooked soybeans in rice straw, which naturally contains the bacteria, and allowing them to ferment. Natto became a staple food in Japanese cuisine, especially during the Edo period (1603–1868 AD), and was commonly consumed in Tokyo [64]. Today, natto remains a beloved and culturally significant food in Japan, often enjoyed as a breakfast dish over steamed rice with condiments like soy sauce, mustard, and green onions. Despite its strong flavor and slimy texture, natto’s nutritional benefits, including its high protein content and probiotic properties, have solidified its place in Japanese culinary heritage [65].

#### 2.1.3. Tempeh

Tempeh’s history is deeply rooted in Indonesian culture and dates back several centuries. This traditional fermented soybean product is believed to have originated from the island of Java [66], where it has been a dietary staple food for generations. The earliest recorded mention of tempeh dates to 1875. In the Serat Centini manuscript, which originates from Java in the early 19th century, the term “tempeh” appears, for instance, in dishes like “jae santen tempeh” (a dish featuring tempeh with coconut milk) and “kadhale tempe serundeng”. This reference indicates that tempeh has been a well-established staple food among the Javanese people for centuries, particularly in regions like Yogyakarta and Surakarta [67]. Tempeh was accidentally “born” likely when cooked soybeans were left out in warm and humid conditions, allowing naturally occurring microorganisms to ferment the soybeans. Tempeh’s production process involves a starter culture known as “ragi tempeh”, which harbors different filamentous fungal species, among which *Rhizopus oligosporus* [68]. This fungus plays a crucial role in fermentation by binding the soybeans together into a cake-like form and producing enzymes that break down the soybean’s proteins and carbohydrates. Tempeh’s popularity gradually spread beyond Indonesia to other parts of Southeast Asia and eventually to the Western world, where it gained recognition for its nutritional value and usage in vegetarian and vegan diets. In the 20th century, tempeh was introduced to the United States, primarily by individuals interested in plant-based diets and alternative protein sources [69].

Today, tempeh is enjoyed worldwide and is celebrated for its nutty flavor, firm texture, and high protein content. It can be grilled, sautéed, stir-fried, or used in a variety of dishes, making it a beloved ingredient in both traditional Indonesian cuisine and modern global cuisine.

#### 2.1.4. Fermented Tofu

It is commonly known as “stinky tofu/sofu” and dates back over a thousand years and has earned its place as a beloved and iconic dish in Chinese food culture. According to legend, tofu was discovered by a Chinese prince named Liu An around 164 BC while experimenting with the coagulation of soy milk. The earliest documented mention of tofu appears in the Han dynasty (206 BC–220 AD) writings. Another historical viewpoint mentioned the inception of tofu can be traced to the year 965 AD when it was initially documented by Tao Ku in China within the text “Anecdotes, Simple and Exotic” [70]. In the Ch’ing dynasty (1644–1912 AD), numerous historical accounts emerged regarding the widespread practice of producing fermented tofu using a molding process. Tofu’s production involves soaking, grinding, and boiling soybeans to produce soy milk, which is then coagulated using either magnesium chloride or calcium sulfate to form curds, which are then pressed into solid blocks of tofu [71]. The process of making fermented tofu with this molding technique is detailed in the book “Shi Xian Hong Mi”, which was written during the middle of the Kang-Xi period (1681–1706). Intriguingly, the author noted that red koji was used in its production [72]. Unlike traditional koji, which employs *A. oryzae* as the starter culture, red koji utilizes *Monascus purpureus*, responsible for its distinctive red pigmentation [73]. When applied in tofu fermentation, red koji imparts characteristic earthy, mildly sweet, and tangy flavor profiles, while its red pigments contribute to the unique coloration of the final product. Tofu’s popularity spread from China to neighboring regions, notably reaching Japan and Korea. In Japan, tofu was introduced around the Nara period (710–794 AD). By the Edo period (1603–1868 AD), tofu had become a common food among the general population [66]. Similarly, it became known as “dubu” in Korea. The spread of tofu continued southward into Southeast Asia, where it became a staple in countries such as Vietnam, Thailand, and Indonesia, adapting to local culinary traditions. Over centuries, tofu has evolved from a regional specialty to a global food, celebrated for its versatility and health benefits [74,75].

#### 2.1.5. Douchi

It is a traditional fermented black soybean product with origins in China, which has been produced for over 3000 years [76]. Its presence is evident from the discovery of douchi in Han Tomb No. 1 at the Mawangdui Han Tomb in South-Central China, dating back to 165 BC. However, the earliest documented reference to douchi can be found in Shi Ming (Explanations of Names), during the Han Dynasty period [77]. Douchi was developed as a method of preserving soybeans through fermentation, which not only extended their shelf life but also enhanced their flavor and nutritional value. The process involves fermenting soybeans, added with salt, with different microbial populations, such as *Aspergillus* spp., *Mucor* spp., and bacteria. Among these, the *Aspergillus*-type douchi is the most ancient and enjoys the broadest production; *Aspergillus* spp. imparts a distinctive umami taste and dark color to the beans [78]. While traditional methods of making douchi involved natural fermentation, modern production often incorporates commercial starters with controlled environments and standardized processes to meet commercial demand. This has led to variations in flavor and quality compared to artisanal douchi. Nowadays, it is available in many international markets, and its influence extends beyond China’s borders. It has been integrated into the cuisines of neighboring countries, including Thailand, where it is known as “tao jiow”, and Vietnam, where it is called “tuong” [79].

#### 2.1.6. Dosa and Idli

As mentioned by Tamang [], Dosa, also known as dosai, is a traditional South Indian dish made from a fermented mixture of rice and black gram. Its existence can be traced back to Tamil Nadu in the first century AD, as mentioned in Tamil Sangam literature. Idli, a round, steamed, spongy, savory staple food made from fermented rice and black lentils, was first documented in Kannada literature in 920 AD, in the work “Vaddaradhane” by Shivakotiacharya. Dosa is thought to have evolved from idli batter. It is believed that the leftover idli batter was spread thin on griddles and cooked to create dosa. Both idli and dosa are highly nutritious and easily digestible due to the fermentation [80], making them staples in South Indian cuisine.

### 2.2. Fermented Dairy Products

The historical roots of fermented dairy products are deeply embedded in diverse cultural traditions worldwide. Milk fermentation, with origins dating back over 10,000 years, likely emerged as a means of preserving dairy products prior to the advent of refrigeration. As noted by Leonardi et al. [81], the domestication of animals began in the Middle Euphrates valley around 11,000 BC for sheep and goats and approximately 10,500 BC for cows. Archeozoological evidence from the mid-9th millennium BC illustrates shifts in the slaughtering practices of these animals [82], marking the onset of their domestication, with this transition occurring at varying times in different regions [83]. Furthermore, evidence of milk fermentation can be traced back to ancient Egypt’s Ptolemaic era, as documented in depictions on stelae, hieroglyphics, and engravings. During this period, milk was stored in egg-shaped earthenware containers sealed with grass to protect it from insects, and was typically consumed shortly after milking [84]. These historical developments led to diverse methods of fermenting milk, resulting in a wide array of fermented dairy products, such as cheese and fermented dairy beverages. These products have played vital roles in regional cuisines, offering distinctive flavors and enhanced nutritional value.

#### 2.2.1. Cheese

Cheese making is believed to have originated around 7000 years ago, likely as a way to store surplus milk and make it more palatable [85,86]. The earliest cheese production can be traced to Southwest Asia and parts of Europe by the late Neolithic, though an earlier origin is also possible [82]. Cheese evolved in the ‘Fertile Crescent’ between the Tigris and Euphratres rivers, in Iraq, some 8000 years ago during the “Agricultural Revolution”, when certain plants and animals were domesticated [83]. Ancient evidence, including clay pots with small holes, suggests that early cheese makers separated curds from whey by allowing fermentation to take place in these containers. The practice gradually spread across the Mediterranean, with the Egyptians, Greeks, and Romans all contributing to the development and refinement of cheese-making techniques. The Roman Empire played a crucial role in disseminating cheese throughout Europe [84], as soldiers and settlers carried the knowledge of cheese production with them. Afterwards, in medieval Europe monasteries became centers of cheese-making expertise. Cheese continued to evolve with the expansion of European exploration and colonization, as cheese-making practices adapted to local ingredients and conditions. In the New World, American colonists created their own cheeses, like cheddar, which gained international recognition. The industrial revolution in the 19th century made cheese production more efficient and cheese became accessible to a wider number of people. This era saw the birth of iconic cheese varieties like Swiss Emmental and Gouda. Today, cheese boasts an astonishing variety of styles, flavors, and textures, with thousands of different varieties produced worldwide. It holds a prominent place in global cuisine, from the rich and creamy Brie of France to the pungent blue cheeses of Roquefort, and from the crumbly feta of Greece to the sharp cheddars of England and the United States [82].

#### 2.2.2. Fermented Dairy Beverages

The tradition of fermented dairy beverages that ultimately gave rise to today’s yogurt extends back millennia, rooted in the domestication of milk-producing animals in the Near East and Central Asia. Archeological and textual evidence indicates that sour milk-based food items were used by pastoral communities as early as the Neolithic period, when milk from goats, sheep, or cattle was unintentionally fermented during transport and storage in warm climates [87,88]. In this Neolithic context (10,000–15,000 years ago), the domestication of sheep, goats, and, later, cattle in the Middle East coincided with use of fermented milk as food [,[84],[87]]. Early written references to fermented milk appear in Indo-Aryan Ayurvedic texts dating to approximately 6000 BCE and among Turkic nomads in Central Asia, where fermenting milk allowed people to benefit from nutrients in milk during transport [87]. The term “yogurt” itself derives from the Turkish verb *yoğurmak*, meaning “to thicken, curdle, or coagulate”, underscoring the product’s cultural origins in Turkish and surrounding societies [87,89]. With the development of microbiology in the late nineteenth and early twentieth centuries, the microbial basis of yogurt fermentation became better understood. In 1905, Stamen Grigorov isolated the bacterium now known as *Lactobacillus delbrueckii* subsp. *bulgaricus* from Bulgarian fermented milk, a discovery that paved the way for the industrial use of defined starter cultures [90].

The historical roots of kefir extend back to a time before written records. It is believed the original kefir grains were gifted by the prophet Muhammad to Orthodox Christians living in Georgia’s Caucasus region [91], possibly in present-day Russia, Armenia, or Georgia. These grains, containing mixed microbial populations, passed down through generations, were used in homes to ferment either cows’ or goats’ milk, typically in leather sacks or oak barrels, and to obtain kefir [91,92]. Much like the widespread popularity of yogurt, kefir gradually spread from the Georgian mountains and is now well-known in Eastern Europe and beyond [93,94,95].

Dahi holds great cultural and culinary significance in South Asia. Its history is closely intertwined with the broader history of dairy farming in the Indian subcontinent. The domestication of cows and buffaloes in ancient India provided a steady supply of milk, which, when fermented into dahi, could be stored and consumed over extended periods without spoiling. This made dahi an essential part of the diet, particularly in areas with hot and tropical climates. Throughout history, the art of dahi making has been passed down through generations, with each region and community developing its unique methods and variations. Dahi is also a key component of religious rituals in Hinduism, often offered as prasad (sacred food) in temples. In recent years, dahi has gained international popularity for its probiotic properties and health benefits.

### 2.3. Fermented Horticultural Produce

Vegetable fermentations were described as early as the Song dynasty in the 10th to 13th centuries AD in China [54]. In Asia and the Far East, a wide range of fermented products were traditionally made using cabbage, turnips, radishes, and carrots. This technology gradually spread to Europe during the 16th century, where regional vegetables like European round cabbages were utilized as the primary raw materials. As European settlers migrated to the New World, they brought along cabbages and knowledge of sauerkraut production. Additionally, it is highly likely that other fermented vegetables, particularly pickles and olives, were produced and consumed in the Middle East since biblical times [8].

#### Kimchi

Kimchi is Korean-style fermented cabbage that can be traced back to the primitive pottery age, where withered vegetables were naturally fermented in seawater [96]. According to historical records, kimchi was invented approximately 4000 years ago, as documented in the Sikyung, a book of ancient Chinese poetry that was compiled during the Zhou dynasty (1046–256 BC). On the other hand, evidence from the Samkuksaki, a historical record of the three states of Korea (Goguryeo, Baekje, and Silla) that was written during the Goryeo dynasty (918–1392 AD), indicates that people were already consuming kimchi in the three states around 1500 years ago. These ancestors discovered the fermentation of kimchi and realized that the presence of red pepper inhibits the growth of harmful microorganisms during fermentation, while promoting the growth of beneficial bacteria, such as LAB [97]. It became closely tied to Korean identity and cuisine during this time. Its role extended beyond just a food preservation method, as it became a staple in Korean households and was incorporated into various dishes. Kimchi making even became a communal activity, with families and neighbors coming together to prepare large batches for the winter months. In the modern era, kimchi has evolved further, with regional variations [98]. It has evolved beyond its traditional role as a side dish and is now featured as a primary ingredient in various Korean dishes. Examples include kimchi fried rice (kimchi bokkeumbap), kimchi noodles (kimchimari guksu), kimchi pancakes, and kimchi dumplings (kimchi mandu) [99].

### 2.4. Fermented Beverages of Vegetable and Fruit Origin

Fermented beverages have been crafted and consumed for millennia, with evidence dating back to ancient civilizations in Mesopotamia, China, and Egypt. These early brewers and fermenters discovered that allowing natural microorganisms, such as yeast and bacteria, to interact with ingredients like grains, fruits, and honey could transform them into flavorful liqueurs or alcoholic beverages. Over time, these discoveries gave rise to a rich tapestry of fermented beverages across the globe, from beer and wine to mead and various indigenous brews. These beverages hold cultural, religious, and social significance, and the techniques and recipes have been passed down through generations, shaping the diverse world of fermented drinks we enjoy today. Although beer and wine are significant examples of fermented beverages, they are not specifically addressed in this review. Readers interested in these products are referred to recent comprehensive reviews [100,101].

#### 2.4.1. Kombucha

Kombucha is defined as ‘a gelatinous mass of symbiotic bacteria (as *Acetobacter xylinum*) and yeasts (as of the genera *Brettanomyces* and *Saccharomyces*) grown to produce a fermented beverage held to confer health benefits’ [102]. The exact origins of kombucha in believed to be in Manchuria region, Northern China [103]. It was known of during the Qin Dynasty, over 2000 years ago, for its detoxifying properties [104,105]. According to legend, Emperor Qin Shi Huang, the first emperor of unified China, was fascinated by the health benefits of tea and was keen to explore its potential, leading to the discovery of kombucha [106]. In traditional Chinese medicine it was believed that fermented teas possessed unique healing properties. Traditional kombucha in China was made by fermenting sweetened tea with a unique Symbiotic Culture Of Bacteria and Yeast (SCOBY). The SCOBY is often referred to as the “tea fungus” or “tea mushroom” due to its appearance [107]. When the SCOBY is inoculated in sweetened tea, bacteria and yeasts start to grow on the surface, conferring a gelatinous aspect to kombucha. Over time, knowledge of kombucha and its fermentation spread throughout China, and it became a staple in traditional Chinese medicine. In 414 AD, the physician Kombu introduced the tea fungus to Japan, using it to alleviate Emperor Inkyo’s digestive diseases. Japanese warriors often carried the invigorating beverage with them onto the battlefield to stay refreshed and strong [108]. With the expansion of trade routes, kombucha, formerly known as “Mo-Gu”, made its way into Russia under various names like Cainiigrib, Cainii kvass, Japonskigrib, Kambucha, and Jsakvasska. Subsequently, it entered other Eastern European regions, including Germany, where it became known as Heldenpilz and Kombuchaschwamm, around the early 20th century [102,105]. In recent years, kombucha has gained global popularity as a functional beverage, appreciated for its unique, tangy flavor [109].

#### 2.4.2. Vinegar

Vinegar is an acidic solution produced through the transformation of ethanol and oxygen into acetic acid (also known as ethanoic acid) and water, by acetic acid bacteria (AAB) [110]. It is an essential ingredient in European, Asian, and various cuisines with a long history dating back to ancient times. Its name originates from the Old French term “vin agre”, signifying “sour wine”. For millennia, vinegar has been both produced and utilized, with archeological evidence of its presence dating back to around 3000 BC in Egyptian artifacts [111]. The Babylonians were known to produce and trade vinegars infused with fruit, honey, and malt up until the 6th century BC Historical accounts, including references in the Old Testament and the writings of Hippocrates, suggest that vinegar was employed for medicinal purposes, particularly in treating and cleansing wounds. In 10th-century China, Sung Tse, regarded as a pioneer of forensic medicine, advocated the use of a mixture of sulfur and vinegar as a handwashing solution to prevent infection and maintain hygiene [112]. It has been an integral part of human diets serving various purposes as a preservative, condiment, aromatizer, and even a medicinal product [113]. Traditionally, vinegar was primarily derived from cereals [114]. Typically, vinegar classification depends on the raw material used in its production. Depending on the substrate, vinegar may be derived from wine, fruit, cereals or other grains, malt or beer, sugarcane (or other sugar-/starch-rich sources), and various plant matrices. Other raw materials can also be used, such as whey or honey, resulting in a wide variety of vinegar types worldwide [115,116]. Among the various types of vinegar, cider vinegar is produced by the acetification of cider, an alcoholic beverage derived from the fermentation of apple juice, and is valued for its health-promoting properties; particularly, it is believed have beneficial effects in controlling blood glucose indices and lipid profile in patients with type 2 diabetes with daily consumption [117]. The specific raw materials, fermentation methods, and microbial communities involved in production result in vinegars with distinct sensory attributes, including unique flavors and aromas.

### 2.5. Fermented Seafood

Seafood fermentation can occur naturally (uncontrolled) or be a controlled process where starter cultures are introduced into the raw material, such as marine fish, shellfish, and crustaceans. Across different regions and cultures, the practice of fermenting seafood dates back thousands of years [118]. It originated as a practical solution to extend in time the availability of fish. Early societies discovered that allowing fish to naturally undergo microbial fermentation not only extended its shelf life but also transformed its texture and flavor. This process was especially vital in coastal and fishing communities, where access to fresh fish could be limited in some seasons of year. Archeological evidence suggests that fish fermentation practices were already in use in ancient Egypt around 3000 BC, where salted and fermented fish were staples in the diet. Similarly, ancient Greeks and Romans employed fermentation techniques to produce garum, a fermented fish sauce highly prized for its savory flavor [119]. In Asia, particularly in countries like China, Japan, Korea, and Southeast Asia, fish fermentation has been a cornerstone of culinary traditions for millennia. In Japan, for example, fermented fish products, like narezushi, date back to the Yayoi period (300 BC–300 AD) [120]. Over time, various methods and techniques for fermenting fish were developed, often incorporating local ingredients and flavors.

#### 2.5.1. Surströmming

Surströmming is a traditional Swedish fermented fish (herring), whose origin is traced to the northern regions of Sweden where preserving fish was essential due to long, harsh winters, causing a scarce supply of fresh food. The name surströmming derives from “sur” (sour or acidic) and “strømming” (the local name for Baltic herring) [121]. This method evolved from packing fresh herring in barrels with minimal salt to ferment. The characteristic pungent odor comes from the action of LAB and enzymes breaking down fish proteins and fats, producing compounds like butyric acid. Historically, the process became popular during periods of trade embargoes and limited salt supply in the 1520s and 1530s, as it allowed for the preservation of large catches using a low amount of salt. In some Swedish regions, surströmming was a staple food and even supplied as army rations in the 17th century [122].

In modern times, surströmming is served with thin bread, potatoes, onions, and sour cream. Fermentation may continue even after canning, causing cans to bulge due to gas production [123].

#### 2.5.2. Nam Pla

Nam pla is a pungent and savory fish sauce, and represents a traditional ingredient in Southeast Asian cuisine, particularly in Thailand, Vietnam, and Laos [124]. Coastal communities in Southeast Asia relied on fishing as a primary source of sustenance. They discovered a natural fermentation by which fish could be transformed into a flavorful and long-lasting condiment. Traditionally, nam pla is made by fermenting small, salted fish, such as anchovies. The fermentation takes place in wooden barrels or clay pots, spanning from several months to several years, depending on regional preferences. During fermentation, naturally occurring enzymes and microorganisms break down the fish proteins into amino acids and produce a liquid that is rich in umami flavor [125]. Nam pla’s history is not confined to its use as a condiment; it has also played a significant role in the development of Southeast Asian cuisine. Today, nam pla remains a vital ingredient in Southeast Asian cuisine and is appreciated for its ability to add depth and complexity to a wide range of dishes.

#### 2.5.3. Feseekh

Feseekh is a traditional Egyptian salted, fermented fish (usual species are: bouri (*Mugil cephalus*), pebbly fish (*Alestes baremoze*), or tiger fish (*Hydrocynus* spp.) that dates back thousands of years [126]. Evidence of fermented fish consumption in the region dates back to at least the time of the Pharaohs. Fish, including mullet, held significant symbolism in ancient Egyptian culture, often associated with fertility and rebirth due to their connection with the Nile River, which was central to Egyptian life. Feseekh is typically made by layering fish with generous amounts of salt in a glass jar. Once layered, the jar is tightly sealed and left to ferment for approximately two months at room temperature, allowing naturally occurring LAB and enzymes to break down the proteins and modify the fish odor and taste [127]. Historically, feseekh was consumed during the springtime celebrations of Sham El Nessim, an Egyptian holiday that dates back to more than 4500 years ago [128,129]. This holiday marks the beginning of spring and the awakening of nature, and eating feseekh became a customary way to celebrate the season. Despite the health concerns associated with improper fermentation and the potential risk of foodborne illnesses, feseekh consumption has not declined in recent years. Today, feseekh remains a part of Egyptian culinary heritage with some modern adaptations and safety precautions.

### 2.6. Fermented Meat

Meat fermentation was used by the ancient civilizations of Northern Europe and Asia in order to use meat as source of nutrients in periods (e.g., long winter) of scarcity with regard to fresh meat. Evidence from ancient Egypt suggests that the fermentation of meat was likely used early on [130].

The earliest records of sausage production trace back to the Sumerians, while written references to these techniques date back to around 600 BC in ancient Greece. Cato the Elder provided detailed descriptions of meat curing (involving, although unconsciously, fermentation) around 160 BC in his work, “De Agri Cultura.” [131]. The Romans possibly adopted these methods from the Gauls and Celts. Celtic roundhouses featured spaces for producing raw dry-cured ham and similar products. The Romans adapted these techniques, incorporating them into their own culinary traditions. Sausage-shaped fermented meats, for instance, are attributed to the Romans, who learned from the Lucanians (southern Italy), the originators of the Greek and Spanish names for dry-cured sausage (“loukaniko” and “longaniza”). The term “salami” likely originates from the Medieval Latin word “salumen” or possibly from the Cypriot city of Salamis. Fermented sausages were known as “brig” in the Roman Empire, a Celtic word meaning “hill”, and possibly the origin of “Brianza” (as in “salame Brianza”) [132]. The Romans laid the foundation for the wide variety of fermented meats in Italy and may have discovered the reddening effect of curing salt, attributed to nitrogen monoxide formation from nitrite, derived from reduction of nitrates (acted by bacteria) in saltpeter. Their advancements in meat-curing techniques contributed significantly to the development of fermented meats in Europe and beyond. With the advent of modern food preservation techniques, such as refrigeration and vacuum sealing, the necessity of fermenting meat declined. However, fermented meat products remain popular in many culinary traditions around the world for their unique flavors and textures. Today, artisanal producers and chefs continue to explore and innovate with traditional fermentation techniques, creating a wide range of fermented meat products enjoyed by people globally.

## 3. Microbial Mechanisms Within Natural Starters Affecting Food Quality: Sourdough as a Paradigm

The transformation of raw materials into fermented foods is underpinned by intricate processes whereby microorganisms, such as LAB and yeasts, exert profound effects on food quality through acidification, proteolysis, and the release of aroma compounds. Overall in those food fermentations driven by LAB —typically belonging to genera such as *Lactobacillus*, *Lactococcus*, *Leuconostoc*, and *Pediococcus*—rapidly metabolize available carbohydrates (with a preference for mono- and di-saccharides) to lactic acid and, for those that display hetero-fermentative metabolism, also to acetic acid or ethanol and carbon dioxide, thereby lowering the pH, which influences texture and flavor, potentially increasing the activity of some food enzymes (e.g., endogenous proteases), and selecting for acid-tolerant microorganisms, such as yeasts [133,134]. By reducing pH and generating organic acids, LAB limit the growth of spoilage and pathogenic microbes [135]. At the same time, many LAB possess a complete proteolytic system. It consists of the following: extracellular or cell-wall associated proteinases that degrade large proteins into peptides; specific transport systems that bring peptides into the cell; and intracellular peptidases that further hydrolyze them into smaller peptides and free amino acids [136,137]. This proteolytic activity contributes not only to microbial growth (providing essential amino acids also in substrates that are poor in free amino acids) but also to flavor and texture development. Indeed, peptides and free amino acids released upon proteolysis act as precursors for volatile organic compounds; in addition, the protein matrix of food is modified, resulting in its softening [138,139]. Furthermore, secondary pathways of LAB metabolism can liberate compounds such as diacetyl/acetoin, hydrogen peroxide, bacteriocins, and exopolysaccharides, which affect sensory aspects, texture, aroma, and the shelf life of food.

Sourdough represents a paradigmatic natural starter culture wherein interactions between populations of LAB and yeasts greatly impact on the quality of food items, namely leavened baked goods. Although different types of sourdough are available, the most intriguing interactions occur in type 1 (*alias* traditional) sourdough, produced and propagated through backslopping [24]. In food items fermented with natural starters, yeasts—commonly from genera such as *Saccharomyces* and *Kazachstania*—may modulate redox and sugar flows and carry out alcoholic fermentation of carbohydrates, producing ethanol and carbon dioxide; in addition, they release a rich suite of flavor-active volatiles via amino acid catabolism (e.g., Ehrlich pathway), ester synthesis, and the conversion of phenolic acids [140,141]. While carbon dioxide results in leavening (for baked goods) or fizz (for beverages), the generated flavor-active volatiles complement the acidifying and proteolytic functions of LAB. In food items fermented with natural starters dominated by LAB and yeasts, those populations may act in sequence or contemporarily.

## 4. The Shift Towards Commercial Starters

Traditionally, food fermentation relied on spontaneous processes, initiated by naturally occurring microorganisms, or was driven by (unconsciously) selected microorganisms inhabiting natural starter preparations, such as sourdough. However, in modern commercial production, many fermented foods have transitioned to fermentation driven by specific starter cultures ensuring standardized quality and minimizing the risk of microbiological hazards. Overall, while artisanal, small-scale processes often utilize natural starters, industrial-scale fermentation predominantly employs commercial starters [142]. The transition towards employing commercial starters in food fermentation processes can be traced back to the late 19th and early 20th centuries [33]. Since the initial advancements made by Emil Christian Hansen at Carlsberg Brewery, who later founded the Christian Hansen company specializing in dairy starter cultures, the starter cultures sector has expanded significantly into a thriving global industry [143]. During this period, several key factors, including population growth and the demand for consistent and varied food products, drove the industrialization of food production [144]. Concurrently, significant advancements in scientific and technological understandings, particularly in microbiology, played a crucial role in the development of commercial starters. In 1857, Louis Pasteur found that lactic fermentation in food was due to bacteria. Milk acidification was found to be caused by a pure culture of *Bacterium lactis*, isolated by Joseph Lister in 1878 [145]. The capacity to obtain pure cultures was a turning point in the production and distribution of starter cultures. Noteworthy figures such as Joseph Harding played a seminal role in advocating for the adoption of standardized starter cultures, catalyzing significant transformations within industries such as dairies [146]. The shift from reliance on natural starters to fermentation methods driven by commercial starters marked a paradigm shift in food production methodologies. Prior to the development of defined starter cultures, fermentation was commonly initiated using the backslopping technique, i.e., the inoculation of the raw material with a small quantity of a previous successfully fermented batch [147]. This practice ensured microbial continuity but posed challenges in standardization and microbial stability. In the 1890s, some food manufacturers began to replace backslopping with powdered starter cultures [148]. Later, advances in biotechnology enabled the direct use of concentrated starter cultures in form of frozen or freeze-dried formulations [149]. Furthermore, Hansen’s pioneering work at the Carlsberg Laboratory in Denmark, which led to the isolation and cultivation of pure yeast strains suitable for brewing, epitomized the pivotal role played by scientific innovation in the shift from natural to commercial starters [150]. The adoption of commercial starters accelerated significantly in the mid-20th century, particularly as food production became more industrialized and mass produced.

Nowadays, many microorganisms are components of commercial starter preparations, marketed to produce various fermented products (Table 2). For example, in cheese making, commercial starters frequently include LAB like *L. lactis* subsp. *lactis* and *Streptococcus thermophilus*, while yogurt production relies on strains of *L. delbrueckii* subsp. *bulgaricus* and *S. thermophilus*. Similarly, fermented vegetables and beverages like sauerkraut, kimchi, beer, wine, and kombucha have shifted to fermentation driven by commercial starters. Leavened baked goods can be produced using either commercial baker’s yeast, a preparation mostly composed of *S. cerevisiae* biomass, characterized by rapid and consistent leavening, or sourdough [151] (Table 2). Although many sourdoughs are produced and propagated at artisanal level and so their microbial composition is undefined, nowadays sourdough starters are also commercially available for professional and home use [24].

In response to increasing consumer awareness and demand for organic and sustainably produced foods, the food industry developed organic starter cultures produced in accordance with organic farming and production standards. Today, organic starters represent a significant niche within the fermentation landscape, offering consumers a choice aligned with their preferences for organic products. The Italian market offers organic starters for fermented foods, such as leavened baked goods (e.g., sourdough), dairy products (e.g., kefir grains), and fermented vegetables (Figure 1). Some of these starters (e.g., “Kefirko organic kombucha starter”) do not report the microbial species, resulting in a hybrid category of “commercial starters with undefined composition”.

The global starter culture market is projected to reach a value of 1.3 billion USD by 2025 [184]. Key contributing countries include China, the United States, South Korea, and Japan. According to the same analysis, the market is expected to grow at a compound annual growth rate of 6.3%, reaching approximately 2.5 billion USD by 2035. Furthermore, bacteria-based starter cultures are anticipated to dominate the market, accounting for 68% of the total revenue share in 2025. These bacterial cultures are primarily composed of LAB, notably species within the genera *Lactobacillus* and *Streptococcus*.

## 5. Starter Production: A Comparative Analysis of Natural and Commercial Starters

### 5.1. Natural Starters

Natural starters, also known as wild or indigenous starters, harbor microorganisms recruited from various sources (environment, raw materials, and processing tools) [185]. Their microbial composition is a reflection of the unique ecological niches that they originate from, encompassing a diverse array of bacteria, yeasts, and molds. The composition of natural starters is influenced by factors such as geographical location, climate, raw materials, and processing techniques, resulting in a vast microbial diversity.

The production of natural starters originates from the traditional method of backslopping. This process intrinsically exerts selective pressures towards microbial community, caused, for instance, by heat treatment (e.g., natural whey milk cultures obtained after cooking cheese curd at 45–55 °C), specific incubation temperatures, and low pH. Unlike more controlled methods, no special precautions are taken to prevent the environmental contamination of natural starters, and there is minimal control during their reproduction (also referred to as “propagation”). Consequently, natural starters consist of undefined mixtures of multiple microbial strains and/or species, and their microbial community often shows time-dependent and almost unpredictable variations (sometimes even from one month to the following month) [186]. The acidification process, driven by naturally occurring bacteria (such as LAB), is often an important driver of the microbial community of natural starters [187]. Acidifying bacteria are predominantly found within the phylum Firmicutes, which includes key LAB genera such as *Lactobacillus*, *Lactococcus*, *Leuconostoc*, *Oenococcus*, *Pediococcus*, *Streptococcus*, *Enterococcus*, *Tetragenococcus*, *Aerococcus*, *Carnobacterium*, *Weissella*, *Alloiococcus*, *Symbiobacterium*, and *Vagococcus* [188]. In addition, some acid-producing bacteria belong to the phylum *Actinobacteria*, notably *Bifidobacterium* and *Atopobium*, which, while not classified as LAB, also contribute to fermentation through carbohydrate metabolism and organic acid production. Sourdough used as a natural starter for bread making harbors LAB (typically rod-shaped and originating from flour and bakery environments) that lower the pH of a wheat/rye-based dough to values ranging from 3.7 to 4.8 [189]. As a result of the inherent variability of natural starters, different flavor profiles and sensory attributes could make them prized assets in artisanal food production [190].

The unique and complex flavor profiles of fermented food obtained through natural starters results from the intricate metabolic activities of the diverse microbial populations harbored in the starter preparation (Table 3). These activities produce a myriad of compounds during fermentation, including organic acids, alcohols, esters, and aromatic compounds, all of which contribute to the sensory characteristics of the final product [191]. Social, economic, and environmental trends are leading to an increasing reliance on fermentations driven by natural starters, particularly within the realm of traditional, regional, and artisanal foods, including those with either the Protected Designation of Origin (PDO) or the Protected Geographical Indication (PGI) status. This shift also extends to organic and biodynamic production methods [192]. Finally, the high bacterial diversity characterizing many natural starters protects against fermentation failure caused by bacteriophages (Table 3).

However, natural starters display some issues. One issue is that, in most cases, they are produced and propagated in the same place (e.g., bakery or dairy farm) where they are used. This implies the presence of some well-educated (especially for traditional sourdough) professionals caring for the correct propagation and performances of the natural starter. In addition, even in presence of well-educated professionals, some variables cannot be timely controlled (e.g., nutrients and microorganisms in the ingredients used for propagation). This implies both a higher risk of undesired microorganisms, detrimental to fermented food safety and shelf life, as well as variability of consistency. The latter aspect is particularly critical in the context of large-scale commercial operations, because the inherent variability of natural starters can make it difficult to achieve consistent product quality and time-reproducible fermentation outcomes (Table 3).

The variability of consistency inherent to natural starter preparations is well-known by those bakers who use traditional sourdough for their leavening baked goods. Figure 2 exemplifies the different outcomes in the reproducibility of food quality due to use of natural starter vs. commercial starter. In most cases, people appreciate the butter aroma related to diacetyl in leavened baked goods. If the microbial community of a type-I sourdough (the natural starter “par excellence” in the field of baked goods) is dominated by *S. cerevisiae* and two LAB species, one of which is a diacetyl producer (such as some strains of *L. lactis*), the resulting final product will have a buttery note. After several backslopping steps, by which this starter is propagated at bakery, using water and flour (intrinsically not sterile), a change may occur in the dominant populations: another species (e.g., *L. plantarum*), possibly originating from the flour used for propagation, may replace the diacetyl producer, resulting in a leavened baked good perceived as less pleasant than before. On the other hand, if the baker uses commercial cultures of *S. cerevisiae* and diacetyl producing *L. lactis*, the resulting baked goods will be characterized, after time, by a buttery note, for as long as the baker uses those commercial starters (Figure 2).

### 5.2. Commercial Starters

In contrast to natural starters, commercial starters are intentionally selected microbial strains, produced under controlled conditions and designed to provide consistent and predictable fermentation outcomes. The latter trait makes them invaluable tools in large-scale food production [162]. The production of commercial starters involves a series of carefully orchestrated steps, from strain isolation and selection to propagation, formulation, and quality control. The journey of a commercial starter begins in the laboratory, where strains are isolated, identified, and selected based on their fermentative capabilities, flavor-producing abilities, and stress tolerance [193]. Selected microbial strains intended for use as starter cultures in the European Union must undergo thorough evaluation and be included in the Qualified Presumption of Safety (QPS) list maintained by the European Food Safety Authority (EFSA). The QPS framework serves as a pre-market safety assessment tool for biological agents used in food and feed, streamlining regulatory approval while ensuring a high safety standard. Assessments are conducted primarily at the species level, based on four main criteria: defined taxonomic identity, a comprehensive body of knowledge supporting safe use (familiarity), intended use, and absence of pathogenicity or harmful metabolite production, including biogenic amines and toxins [194,195]. The presence of acquired antimicrobial resistance genes is also critically evaluated. The QPS list, updated annually, includes a range of LAB (e.g., *Lactobacillus* and *Lactococcus*), certain *Bacillus* species with specific safety qualifications, and yeasts such as *S. cerevisiae*, though some subtypes require caution in vulnerable populations. Filamentous fungi remain excluded due to ongoing taxonomic revisions and limited toxicological data. Only microbial strains meeting these stringent criteria are recommended as food ingredients, thereby ensuring consumer safety and consistency in fermentation processes.

Once selected, the strain is propagated (cultivated) under sterile conditions in controlled tanks filled with nutrient-rich media optimized for obtaining the highest yield in biomass. The large-scale production of a strain that will be part of a commercial starter is obtained using bioreactors equipped with precise control systems for temperature, pH, and oxygen levels [196]. The microbial biomass obtained after cultivation is harvested, concentrated, and formulated into various forms, such as freeze-dried powders, liquid cultures, or granules, which are easy to handle and store. The facilities where commercial starter cultures are produced adhere to Good Manufacturing Practices (GMP) and other regulatory standards to ensure the safety, purity, and efficacy of the starter cultures. This ensures that the production of commercial starters adheres to the Hazard Analysis and Critical Control Point (HACCP) system. Quality assurance procedures, including microbial testing, potency assays, and stability studies, are conducted to verify the performance and shelf life of the commercial starters before they are released to the market [197]. A given commercial starter may harbor one or more microbial species, each one individually propagated and then combined to give a defined single (one species) or mixed (two or more species/strains) starter culture.

Besides being particularly precious in large-scale food process plants, defined starter cultures enhance the predictability of small-scale fermentation processes, improve the aroma of traditional products, and increase product safety [143]. Unlike many natural starters, the commercial ones are produced by industries, so that the food manufacturer does not have to care about starter propagation. In addition, most of them can be used by operators with a low to moderate level of education. Their inclusion prevents interruptions in fermentation and the formation of unwanted metabolites [186]. By using specific strains of bacteria and fungi, commercial starters ensure a uniform outcome in terms of flavor and texture. Additionally, they minimize the risk of contamination by undesirable microorganisms, significantly enhancing food safety. Moreover, commercial starters typically accelerate the fermentation. For instance, bread leavening with commercial starters consisting of *S. cerevisiae* is much faster (1–2 h) compared to traditional sourdough (4–8 h). Notwithstanding the level of customization of commercial starters cannot be as high as that of natural starters, commercial starters can be tailored to specific applications to meet the requirements of different food products and processing conditions (Table 3).

Commercial starter cultures also display some disadvantages. One of them is that in most products with PDO or PGI designation, their use is not envisaged in the production disciplinary because they would cause a loss of typicity and traditionality. Furthermore, those commercial starters based on one or few bacterial strains are inherently exposed to infections of bacteriophages, which would cause a rapid loss of metabolic activity, thus impeding the onset of fermentation and, consequently, the obtainment of the fermented food/beverage (Table 3).

## 6. Fermented Food and Beverages from Organic Agriculture: How to Respect Regulation About Organic Food Using Microbial Starters

The production of organic starters involves cultivating microorganisms under strict conditions to meet organic certification standards. Organic starters are essential in various fermentation processes, including dairy products like organic yogurt and cheese, as well as sourdough bread and fermented vegetables. EC Regulation 834/2007 specifies certain products and substances permitted for use in the production of processed organic foods. Some of these substances, like potato starch and vegetable oils, must be of organic origin. However, other substances can derive from conventional farming, provided their use does not exceed 5% of the total production. The regulation specifies that, “For the production of organic yeast, only organically produced substrates shall be used. Additionally, organic yeast must not be included in organic food or feed alongside non-organic yeast”. Despite this, there are no explicit standards or requirements for the production of organic starters, with certification criteria currently only encompassing yeast.

For other microorganisms usable as starters, the future key aspects of organic starters production may include adhering to organic certification requirements, which mandate the use of organic ingredients throughout the propagation. This may involve employing organic substrates/media for microbial growth, such as organic milk for dairy starters or organic flour for sourdough starters. The selection of specific microbial strains is critical, focusing on those naturally occurring or isolated from organic sources to ensure they thrive in organic environments and contribute to desired fermentation characteristics. Organic starter production has the potential to emphasize environmental sustainability by reducing synthetic inputs and supporting the ecological balance within production systems, aligning with organic farming principles to minimize the ecological footprint of fermentation. Overall, organic starter production aims to meet consumer demand for high-quality, organic-certified foods.

Within the context of organic agriculture, the production of fermented products holds particular significance. However, ensuring compliance with regulatory standards related to organic food production presents unique challenges, particularly regarding the use of microbial starter cultures. At the heart of organic fermentation lies a delicate balance between tradition and regulation, where the artistry of fermentation meets the rigor of certification requirements. Organic agriculture is founded on management practices that prioritize the preservation, restoration, maintenance, and enhancement of ecological balance. It adheres to sustainability principles, thereby contributing to environmental, economic, and social sustainability goals [198]. These goals mandate strict adherence to principles such as soil health, biodiversity conservation, and the exclusion of synthetic inputs, including pesticides, fertilizers, and genetically modified organisms (GMOs). The principles extend to all stages of food production and the use of microbial starters in organic fermentation must also align with them. As such, the sourcing and cultivation of microbial starters must adhere to organic certification standards, ensuring that they are derived from organic-certified sources and cultivated using organic-compliant practices. Microbial starters used in organic fermentation must of course meet stringent criteria for purity, safety, and functionality, as well as compatibility with organic substrates and processing conditions.

While organic fermentation prioritizes natural processes and microbial diversity, ensuring microbial safety within these systems remains a regulatory and technological challenge. As mentioned above, organic certification frameworks, such as EU Regulation 2018/848, only mandate that all microbial cultures used in organic food processing be non-GMO, produced without synthetic growth media, and free from antibiotics or chemical preservatives. However, these regulations primarily emphasize input integrity rather than detailed microbiological risk assessment. As a result, when natural or undefined starters are used, variability in microbial composition can increase the risk of contamination with pathogenic species. To mitigate that risk, EFSA recommends the use of QPS species and the periodic verification of microbial identity, purity, and absence of virulence or antimicrobial resistance genes [195]. In practice, producers of organic fermented foods are encouraged to implement GMP, HACCP systems, and whole-genome-based microbial traceability to ensure both compliance and consumer safety. Integrating these safety evaluations into organic certification schemes would help bridge the current gap between traditional artisanal practices and modern food safety requirements, thereby strengthening the integrity and market credibility of organic fermented products.

## 7. Embracing Nature: Harnessing the Potential of Natural Starters in Organic Food Processing

Organic food processing is increasingly focusing on the use of natural starters. Similarly to commercial starters, they may benefit nutrition and health, for instance by increasing the bioavailability of vitamins and minerals; additionally, natural starters may include microorganisms with probiotic traits [199].

Natural starters, as well as commercial ones, can play a crucial role in preserving organic foods and ensuring their safety without synthetic preservatives, which aligns with the principles of organic food processing [200]. However, in contrast to commercial starters, the use of natural starters for organic food production targets one of the broader goals of organic agriculture, namely promotion of biodiversity, because, as discussed above, natural starters are characterized by higher microbial diversity than commercial starters. Finally, natural starters, unlike commercial starters, do not require specialized laboratory equipment. Although no generalization can be made, it may be hypothesized that a fermented food obtained by using a natural starter produced at the same place could be more environmentally sustainable than its counterpart, produced with a commercial starter.

## 8. Challenges and Future Perspectives

### 8.1. Overcoming Hurdles: Identifying Limitations and Addressing Challenges in the Use of Natural Starters in Organic Food Processing

Despite the numerous advantages of natural starters, several challenges must be addressed to fully realize their potential in organic food processing. One significant challenge is the variability and unpredictability inherent in natural starter cultures [187,201]. This may be solved via the application of standardized protocols in their production and via the selection of microbial strains that are robust, i.e., capable of controlling contaminant microorganisms that may become dominant in natural starter cultures, thus causing deviations from the standard quality of a given fermented food/beverage. In practice, robust LAB and yeast strains in starter cultures can exert competitive exclusion and secrete antimicrobial compounds, helping to inhibit spoilage organisms and pathogens [202]. Among contaminating microorganisms, some can increase the risk of either foodborne disease or spoilage before the end of shelf life. This challenge must be faced via the selection of bacteriocinogenic microorganisms as components of natural starters and through the application of stringent hygiene practices [203].

In addition to technical challenges, there are some logistical and economic issues that must be considered. The cultivation and maintenance of natural starter cultures may require adequate infrastructure and expertise, which may pose barriers to entry for small-scale producers. Furthermore, the scalability of natural starters’ production may be limited compared to their commercial counterparts, which are produced on an industrial scale using automated processes and standardized protocols [204]. Finding ways to overcome these barriers will be crucial for expanding their adoption in organic food processing. Nevertheless, addressing these challenges requires collaboration between researchers, producers, and regulatory authorities to develop standardized protocols, quality control measures, and safety guidelines for the use of natural starters in organic food processing. Investing in research and innovation to improve our understanding of natural starters’ application and of how to optimize their performance will be essential to overcome hurdles and unlock their full potential in organic food processing.

### 8.2. Innovating for the Future: Advancements and Future

Recent innovations in starter culture production leverage advanced screening techniques and omics technologies to enhance fermentation. High-throughput screening, along with genomics, proteomics, and metabolomics, provides deep insights into microbial strains, enabling the selection of those with enhanced fermentation performance, flavor development, and probiotic traits [205,206,207,208,209,210]. Synthetic biology is emerging as a promising field for future starter culture production. It enables the precise manipulation of microbial genomes, allowing for the engineering of strains with tailored traits and functionalities. These technologies facilitate the identification of specific genes or pathways responsible for beneficial traits, streamlining the selection process. Clustered Regularly Interspaced Short Palindromic Repeats/CRISPR associated protein 9 (CRISPR-Cas9) and other genetic engineering tools have revolutionized starter culture development by allowing the precise editing of microbial genomes. CRISPR-Cas9 has also become a standard practice in many laboratories, including those that perform manipulations with yeast [211]. This technique leverages the natural “adaptive immunity” mechanism observed in bacteria and archaea to create a tool capable of precise genome editing in any organism [212]. Consequently, this enables the enhancement or introduction of traits such as stress tolerance, improved metabolic pathways, and bacteriophage resistance, ensuring consistent and reliable fermentation outcomes.

Under current EU law, most designer microbes intended for food fermentations are treated as GMOs, imposing stringent approval and labeling requirements. In 2018, the Court of Justice of the EU confirmed that organisms modified by new techniques (e.g., CRISPR) qualify as GMOs under Directive 2001/18/EC [213,214]. As a result, any food containing or produced from an engineered microorganism falls under Regulation (EC) No. 1829/2003 requires a full EFSA safety assessment and a pre-market authorization [215,216]. Only if an engineered microbe is used as a processing aid (i.e., completely removed from the final product with no DNA or cells detectable) can the food be considered “produced with” a GMO and thus escape GMO labeling [217,218]. In that case, no separate GMO authorization is needed, though applicants must still demonstrate the absence of viable modified cells or DNA in the finished food. In practice, this means that purified additives (enzymes, vitamins, flavorings, etc.) made by genetically modified microorganism (GMM) fermentation can reach the market without GMO labeling, but live or DNA-containing engineered cultures in foods cannot. For example, the EU now requires that any food enzyme, flavoring, or additive derived from a GMO be authorized through EFSA’s evaluation. Together with lengthy approval timelines, these rules make new engineered starter cultures hard to deploy under today’s framework; regulatory agencies in the EU (as elsewhere) investigate both the modified microbe and its product for safety.

On the other hand, under current EU organic regulations, the use of GMOs, including GMMs or organisms altered by new genomic techniques, is strictly prohibited. EU Regulation 2018/848 (which replaced earlier rules) expressly bans the use of “GMOs and products produced from or by GMOs” in organic production. This prohibition covers not only plants but also microorganisms used in any part of the organic food chain (e.g., processing aids, enzymes, starter cultures) since microorganisms fall under the category of “microorganism or animal material” derived from or by GMOs. Organic rules also require that for processed organic foods, only a limited set of additives, processing aids, enzymes, microorganisms etc., are permitted—but only if they are authorized under organic rules, which exclude those derived from or through GMOs. Thus, even if a microorganism engineered via synthetic biology or CRISPR offers desirable traits (flavor, stress tolerance, etc.), its engineered status disqualifies it for use in organic-certified products. Until the legislation is revised or new guidance is issued, the feasibility of using gene-edited microbes in food remains very limited under EU law.

There are notably some sustainable practices that are being integrated into starter culture production. The use of agro-industrial byproducts as substrates is gaining popularity, reducing production costs and minimizing waste. For instance, whey, a byproduct of cheese production, can be used to cultivate LAB, contributing to a circular economy [219]. Additionally, innovations in starter cultures for plant-based food items cater to the increasing demand for vegan products. These cultures are developed to effectively ferment plant-based substrates, producing high-quality plant-based cheeses, yogurts, and other fermented food items with enhanced texture, flavor, and nutritional profiles. They could also potentially exhibit enhanced growth and metabolism in plant-based milk products, in comparison to dairy strains, due to their broader metabolic capacities [220].

Automation in fermentation has led to the development of efficient and scalable starter culture production systems. Automated systems can monitor and control environmental conditions such as temperature, pH, and oxygen levels in real time, optimizing growth kinetics and biomass yields. Digital twins and predictive modeling further enhance efficiency by simulating fermentation and allowing producers to adjust parameters for optimal outcomes, reducing the need for trial and error [221]. Lastly, advancements in downstream processing technologies, such as freeze drying, spray drying, and encapsulation, have improved the stability, shelf life, and functionality of commercial starter cultures. These techniques allow for the production of starter cultures in convenient, user-friendly formats that are easy to store, transport, and use. Formulation and delivery system innovations, such as microencapsulation and protective coatings, further enhance the viability and survivability of starter cultures during storage.

Innovating for the future involves leveraging advancements in technology, science, and sustainability to enhance the utilization of natural starters in organic food processing. Future advancements may include the development of novel techniques for microbial isolation, characterization, and selection to identify microbial strains with desirable fermentation properties and functional attributes. Additionally, advances in fermentation science and biotechnology may enable the engineering of custom, tailored starter cultures with enhanced functionalities and performance characteristics.

Moreover, advancements in food safety technologies and quality assurance systems can help mitigate food safety risks associated with natural starter cultures, ensuring the stability of fermented products. Implementing blockchain technology and traceability systems can enhance transparency and accountability throughout the supply chain, enabling consumers to make informed choices about the products they purchase.

## 9. Conclusions

The rich microbial diversity of natural starter cultures offers a major advantage for fermented foods and beverages, as microbial interactions during fermentation generate products with distinct—and often unique—sensory traits. This is particularly relevant for organic, fermented foods, whose consumers typically value not only more sustainable production systems, but also sensory profiles clearly differentiated from those of large-scale, industrial products. However, the high biodiversity of natural starters can also lead to time-dependent variations in sensory quality. In contrast, commercial starters facilitate process control and ensure consistent product quality over time, with minimal risk of contamination by undesired microorganisms. Yet they are usually unsuitable for traditional or typical products because they tend to “flatten” sensory profiles, undermining the features that define product typicity.

For producers of fermented foods and beverages, two complementary strategies can help maximize the benefits of natural starters while limiting their inherent risks: (i) implementing control measures and rapid assays during starter propagation to verify the absence of undesired microorganisms, and (ii) improving the training and education of operators responsible for managing natural starters. Enhanced knowledge in this area not only increases awareness of practical measures to reduce variability but also strengthens the ability of operators to adapt starter cultures to evolving consumer expectations.

For starter culture manufacturers, we recommend collaborating with research institutions to develop natural starter cultures that combine high viability with robustness. Once available on the market, such cultures would provide producers with an alternative to self-propagated natural starters and would allow the use of commercial preparations capable of preserving the distinctive sensory qualities of traditional and typical fermented foods and beverages. These recommendations apply to both conventional and organic production systems, but they are particularly relevant in the organic sector, where biodiversity—including microbial diversity—plays a central role.

## Figures and Tables

**Figure 1 foods-14-04240-f001:**
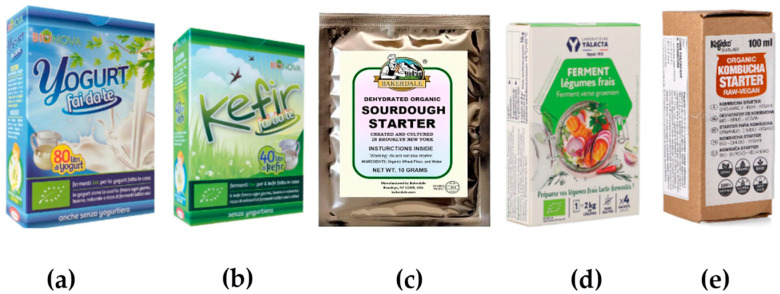
Organic commercial starters in the Italian market; (**a**) Bionova organic yogurt starter (*S. thermophilus* and *L. delbrueckii* subsp. *bulgaricus*); (**b**) Bionova organic kefir starter (Genera: *Lactobacillus*, *Lactococcus*, *Leuconostoc*, *Saccharomyces*); (**c**) Bakerdale organic sourdough starter (microorganisms not specified); (**d**) Yalacta organic vegetable starter (*S. thermophilus*, *L. delbrueckii* subsp. *bulgaricus*, *L. plantarum*, *L. acidophilus*, and *Bifidobacterium lactis*); and (**e**) Kefirko organic kombucha starter (microorganisms not specified).

**Figure 2 foods-14-04240-f002:**
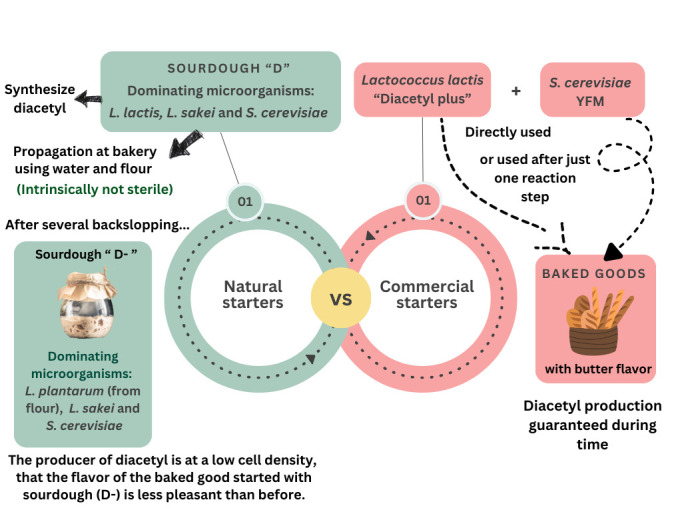
Example of the differences between a natural and a commercial starter, in terms of consistency and change in food quality, attributed to the change in microbial community of sourdough.

**Table 1 foods-14-04240-t001:** Fermented cereal- and legume-based food products from various substrates, with indication of country of origin/common consumption and microorganisms involved in processing [35,38,39,40,41,42,43,44,45,46,47,48,49,50,51].

Country	Product	Substrate	Microorganisms Conducting Fermentation
Albania, Turkey, Bulgaria, and Romania	Boza	Wheat, millet, maize, and other cereals	Bacteria: *Lactobacillaceae* (formerly *Lactobacillus* species) and *Leuconostoc mesenteroides*; yeasts: *Saccharomyces cerevisiae*
Benin, and Togo	Mawè	Maize	Bacteria: *Limosilactobacillus fermentum* (formerly *Lactobacillus fermentum*), *Limosilactobacillus reuteri* (formerly *Lactobacillus reuteri*), *Levilactobacillus brevis* (formerly *Lactobacillus brevis*), *Latilactobacillus curvatus* (formerly *Lactobacillus curvatus*), *Weissella confusa*, *Ligilactobacillus salivarius* (formerly *Lactobacillus salivarius*), *Lactococcus lactis*, *Pediococcus pentosaceus*, *Pediococcus acidilactici*, *L. mesenteroides*; yeasts: *Pichia kudriavzevii* (formerly *Candida krusei*), *Kluyveromyces marxianus* (formerly *Candida kefyr*), *Candida glabrata*, and *S. cerevisiae*
Brazil	Cachiri	Maize	Bacteria: *Lactobacillaceae*; yeasts: *S. cerevisiae* and *Candida* spp.
	Fubá	Germinated and fermented maize grains	Bacteria: *Lactobacillaceae*; yeasts: *Saccharomyces* spp.
Botswana	Bogobe	Sorghum	Bacteria: *L. reuteri*, *L. fermentum*, *Lacticaseibacillus harbinensis* (formerly *Lactobacillus harbinensis*), *Lactiplantibacillus plantarum* (formerly *Lactobacillus plantarum*), *Lentilactobacillus parabuchneri* (formerly *Lactobacillus parabuchneri*), *Lacticaseibacillus casei* (formerly *Lactobacillus casei*), and *Loigolactobacillus coryniformis* (formerly *Lactobacillus coryniformis*)
Burkina Faso	Bikalga	Roselle (*Hibiscus sabdariffa*)	Bacteria: *Bacillus subtilis*, *Bacillus licheniformis*, *Bacillus cereus*, *Bacillus pumilus*, *Pseudobacillus badius* (formerly *Bacillus badius*), *Brevibacillus porteri* (formerly *Bacillus bortelensis*), *Lysinibacillus sphaericus* (formerly *Bacillus sphaericus*), and *Lysinibacillus fusiformis* (formerly *Bacillus fusiformis*)
	Soumbala	Locust bean	Bacteria: *B. subtilis*, *B. pumilus*, *Priestia megaterium* (formerly *Bacillus megaterium*) and *B. licheniformis*
China	Chee-fan	Soybean wheat curd	Molds: *Mucor* spp. and *Eurotium herbariorum* (Formerly *Aspergillus glaucus*)
	Furu/Lufu/Sufu	Soybean curd	Molds: *Actinomucor*, *Mucor*, *Rhizopus*,
	Lao-chao	Rice	*Rhizopus arrhizus* (formerly *Rhizopus oryzae*), *Rhizopus microsporus* var. *chinensis*, and *Saccharomycopsis guttulate* (formerly *Cyniclomyces guttulatus*)
	Yandou	Soybean	*B. subtilis*
China and Taiwan	Douchi	Soybean	Molds: *Mucor racemosus* (*Chlamydomucor racemosus*), *Aspergillus oryzae*, *Aspergillus egyptiacus*, *Rhizopus delemar* (formerly *Rhizopus oryzae*), and *R. microsporus* var. *oligosporus*
	Meitauza	Soybean	*B. subtilis*
China, Taiwan, Thailand, and Philippines	Ang-kak	Red rice	Molds: *Monascus purpureus*
Congo	Poto poto	Maize	Bacteria: *L. mesenteroides*
Colombia	Champuz	Maize or rice	*Lactobacillus* spp. and *S. cerevisiae*
Cyprus, Greece, and Turkey	Tarhana	Sheep milk, wheat	Bacteria: *Lactobacillus delbrueckii* subsp. *bulgaricus*, *Streptococcus thermophilus*, *L. lactis*, *Lactobacillus acidophilus*, *L. mesenteroides* subsp. *cremoris*, and *L. casei*; yeasts: *S. cerevisiae*
Egypt	Kishk	Oat, barley, bulgur, wheat	Bacteria: *L. plantarum*, *L. brevis*, *L. casei*, and *B. subtilis*
	Sourdough	Rye, wheat	Bacteria: *Fructilactobacillus sanfranciscensis*, *L. plantarum*, *Lacticaseibacillus paracasei *(formerly *Lactobacillus paracasei),* *L. casei*, *Pediococcus pentosaceus*, *L. mesenteroides*, *Weissella cibaria*, and *W. confusa*; yeasts: *S. cerevisiae*, *Kazachstania humilis*, *Kazachstania exigua*, *Wickerhamomyces anomalus*, *Torulaspora delbrueckii*, and *P. kudriavzevii*
Ethiopia	Enjera/Injera	Tef flour, wheat	Bacteria: *L. delbrueckii* subsp. *bulgaricus*; yeasts: *Meyerozyma guilliermondii* (formerly *Candida guilliermondii*)
	Borde	Maize, sorghum, wheat, finger millet, and barley	Bacteria: *Lactobacillaceae* and *Leuconostoc* spp.; yeasts: *S. cerevisiae* and *Candida* spp.,
Ghana	Banku	Maize, or maize and cassava	*Lactobacillus* spp.
Ghana	Kenkey	Maize	Bacteria: *L. fermentum*, *L. reuteri*, *L. plantarum*, *L. brevis*, *P. pentosaceus*, *L. mesenteroides*, and *P. acidilactici*; yeasts: *S. cerevisiae*, *P. kudriavzevii* and *Scytalidium candidum*
	Koko	Maize	Bacteria: *Klebsiella aerogenes* (formerly *Enterobacter cloacae*), *Acinetobacter* spp., *L. plantarum*, and *L. brevis*; yeasts: *S. cerevisiae* and *Candida vini* (formerly *Candida mycoderma*)
Ghana, and Nigeria	Dawadawa	Locust bean	*Bacillus* spp.
India	Adai	Cereals/legume	Bacteria: *Pediococcus* spp., *Streptococcus* spp., and *Leuconostoc* spp.
	Ambil	Ragi flour, cooked rice, and buttermilk	Bacteria: *Lactobacillus* and *Streptococcus* species
	Anarshe	Rice	*Lactobacillus* spp.
	Bekang	Soybean	*Bacillus* spp.
	Bhallae	Black gram	Yeasts: *Candida famata* (formerly *Debaryomyces hansenii*), *Cutaneotrichosporon curvatum* (formerly *Candida curvata*), *Tausonia pullulans* (formerly *Trichosporon pullulans*), *Ogataea polymorpha* (formerly *Hansenula polymorpha*), and *Candida parapsilosis.* Bacteria: *L. fermentum* and *L. mesenteroides.*
	Dhokla	Bengal gram (*Cicer arietinum*), dehulled black gram (*Phaseolus mungo*), and milled rice	Bacteria: *L. fermentum* and *L. mesenteroides*; mold: *Wynnella silvicola* (formerly *Helvella silvicola*)
	Hawaijar	Soybean	*Bacillus* spp.
	Tungrymbai	Soybean	*Bacillus* spp.
	Vada	Cereal/legume	Bacteria: *Pediococcus*, *Streptococcus* and *Leuconostoc* spp.
India and Himalaya	Jaanr	Millet	Yeasts: *W. anomalus* (formerly *Hansenula anomala*), and *Mucor indicus* (formerly *Mucor rouxianus*)
India, Nepal, and Bhutan	Kinema	Soybean	Bacteria: *B. subtilis*, *B. licheniformis*, *B. cereus*, *Niallia circulans* (formerly *Bacillus circulans*), *Bacillus thuringiensis*, *L. sphaericus*, and *Enterococcus faecium*; yeasts: *C. parapsilosis* and *S. candidum*
India, Nepal, and Pakistan	Jalebi	Wheat flour	Yeasts: *Saccharomyces uvarum* (formerly *Saccharomyces bayanus*)
India and Pakistan	Rabadi	Buffalo or cow milk and cereals, pulses	Bacteria: *Bacillus* and *Micrococcus* spp.; molds: *Talaromyces acidilactici* (formerly *Penicillium acidilactici*),
India and Sikkim	Bhattejaanr	Rice	Molds: *Rhizopus arrhizus*; Yeasts: *W. anomalus*
India, Sri Lanka, Malaysia, Singapore	Dosa	Rice and black gram or other dehusked pulses	Bacteria: *L. mesenteroides*, *Streptococcus faecalis*, and *L. fermentum*; yeasts: *Tausonia pullulans* (formerly *Tricholsporon pullulans*), *Torulopsis* (now classified under *Candida* and *Cryptococcus*), and *Candida* spp.
	Idli	Rice, black gram or other dehusked pulses	Bacteria: *Bacillus amyloliquefaciens* (now recognized as *Bacillus velezensis*), *L. mesenteroides*, *L. plantarum* subsp. *plantarum*, *Lactiplantibacillus pentosus *(formerly *Lactobacillus pentosus*), *L. lactis*, and *Lactiplantibacillus argentoratensis* (formerly *Lactobacillus plantarum* subsp. *argentoratensis*); yeasts: *Torulopsis*, *Candida*, *T. pullulans*, and *Pediococcus cerevisiae* (now named as *Pediococcus damnosus* or *Pediococcus acidilactici*)
Indonesia	Brem	Cassava, glutinous rice	Bacteria: *Acetobacter aceti*; yeasts: *S. cerevisiae*; molds: *R. delemar*, *Monilesaurus rouxii*, and *A. oryzae*,
	Brembali	Rice	Yeasts and Molds: *M. indicus* and *Candida* spp.
	Kecap	Soybean, wheat	Bacteria: *Tetragenococcus halophilus* (formerly *Pediococcus halophilus*)
	Oncom Hitam (Black Oncom) and Oncom Merah (Orange Oncom)	Peanut press cake, tapioca, soybean curd starter	Molds: *Chrysonilia sitophila* (*Neurospora sitophila*) and *Rhizopus microsporus* var. *oligosporus*
	Tape Ketan	Glutinous rice, Ragi	LAB; Yeasts: *S. cerevisiae*, *W. anomalus*, *C. glabrata*, *C. tropicalis*, *Hanseniaspora uvarum*, and *Pichia kudriavzevii*; molds: (from starter “ragi tape”) *Amylomyces rouxii*, *Rhizopus oryzae*, *M. indicus*, and *A. oryzae*.
	Tauco	Soybean	Bacteria: *L. delbrueckii*; yeasts and molds: *Hansenula* spp., *Zygosaccharomyces* spp., and *A. oryzae*
Japan	Miso	Soybean	Molds: *A. oryzae*, *Clavispora lustaniae*, and *M. guilliermondii*; Bacteria: *B. velezensis*, *B. subtilis*, *Enterococcus durans*, *Rothia kristinae* (formerly *Kocuria kristinae*), *L. plantarum*, *Leuconstoc citreum*, *Leuconostoc pesudomesenteroids*, *P. acidolactici*, *P. pentosaceus*, *W. cibaria*, and *W. confusa*
	Natto	Soybean	Bacteria: *B. subtilis*
Kenya, Uganda, Tanzania	Uji	Maize, sorghum, millet, cassava flour	Bacteria: *L. mesenteriodes* and *L. plantarum*
Korea	Chongju	Rice	Yeasts: *S. cerevisiae*
	Chungkokjang (or jeonkukjang, cheonggukjang	Soybean	Bacteria: *B. subtilus*
	Doenjang	Soybean	Bacteria: *Bacillus*, *Tetragenococcus*, *Staphylococcus*, *Enterococcus*, *Pediococcus*, *Weissella*, *Hyphopichia*, *Debaryomyces*, and *Wickerhamomyces* spp.
	Gochujang	Soybean, red pepper	*B. velezensis*
	Ganjang	Soybean, meju, salt, water	Bacteria: *Cobetia*, *Bacillus*, and *Chromohalobacter* spp.
Mexico	Pozol	Maize	LAB (mainly *L. plantarum*, *L. casei*, and *L. delbrueckii*)
Mongolia	Darassum	Millet	LAB (not specified)
Nepal, India	Maseura	Black gram	Bacteria: *L. fermentum*, *L. salivarius*, *P. pentosaceus*, and *Enterococcus durans*; yeasts: *S. cerevisiae.*
Nigeria	Kunu-zaki	Maize, sorghum, millet	Bacteria: *Corynebacterium*, *Aerobacter*, *Lactobacillus*, *Pediococcus*, *Lactococcus*, *Leuconostoc*, and *Bifidobacterium* spp.
	Ogi	Maize, sorghum, millet	Bacteria: *L. plantarum*; yeasts and molds: *S. cerevisiae*, *Kregervanrija fluxuum* (formerly *C. mycoderma*), *Rhodotorula*, *Cephalosporium*, *Fusarium*, *Aspergillus*, and *Penicillium* spp.
	Okpehe	Seeds from *Prosopis africana*	Bacteria: *Bacillus* spp.
	Ugba	African oil bean (*Pentaclethra macrophylla*)	Bacteria: *Bacillus* spp.
Nigeria, Benin	Iru	Locust bean	Bacteria: *Staphylococcus* and *Bacillus* spp.
	Burukutu	Sorghum	Bacteria: *Leuconostoc* spp.; yeasts: *S. cerevisiae*
Nigeria, Ghana	Busaa	Maize	Bacteria: *Lactobacillus helveticus*, *L. salivarius*, *L. casei*, *L. brevis*, *L. plantarum*, and *Lentilactobacillus buchneri* (formerly *Lactobacillus buchneri*); yeasts and molds: *S. cerevisiae* and *Penicillium damnosus*
Northern India	Dhokla	Rice or wheat and bengal gram	Bacteria: *L. mesenteroides*, *Enterococcus faecalis*; yeasts and molds: *Candida* spp. and *Trichosporon pullulans* (recently known as *Guehomyces pullulans*)
Peru	Chicha	Maize	*Aspergillus* and *Penicillium* spp., yeasts and other bacteria (unspecified)
Sierra Leone	Kinda	Locust bean	*Bacillus* spp.
Southern Mexico	Atole	Maize	LAB (not specified)
South Africa	Mahewu	Maize meal and wheat flour	Bacteria: *L. plantarum* and *L. mesenteroides*; yeasts: *S. cerevisiae*
Sudan	Kawal	Leaves of legume (*Cassia* sp.)	Bacteria: *B. subtilis* and *Propionibacterium* spp.
	Kisra	Sorghum	Bacteria: *Pediococcus pentosaceus*, *W. confusa*, *L. fermentum* (formerly *Lactobacillus cellobiosus*), *L. brevis*, *Lactobacillus amylovorus*, and *L. reuteri*; yeasts: *Sungouiella intermedia* (formerly *Candida intermedia*), *C. famata*, and *S. cerevisiae*
	Merissa	Sorghum and millet	Yeasts: *Saccharomyces* spp.
Syria, Turkestan	Busa	Rice or millet	Bacteria: *Lactobacillus* spp.; yeasts: *Saccharomyces* spp.
Tanzania	Togwa	Cassava, maize, sorghum, millet	Bacteria: *L. plantarum*, *L. brevis*, *L. fermentum*, *P. pentosaceus*, *W. confusa*; yeasts: *P. kudriavzevii* (formerly *Issatchenkia orientalis*), *S. cerevisiae*, *Candida pelliculosa*, and *Candida tropicalis*
Thailand	Khamak (Kao-mak)	Glutinous rice, Look-pang (starter)	Yeasts and molds: *Rhizopus*, *Mucor*, *Saccharomyces*, and *Hansenula* spp.
	Thua nao	Soybean	Bacteria: *B. subtilis*
West Africa	Pito	Maize, sorghum	Bacteria: *Lactobacillus* spp.; yeasts: *S. candidum*, and *Candida* spp.
West, East and Central Africa	Ogiri/Ogili	Melon seeds, castor oil seeds, pumpkin bean, sesame	Bacteria: *B. subtilis*
Western Ughanda	Bushera	Germinated sorghum and millet grains	Bacteria: *L. brevis*, *Lactobacillus*, *Lactococcus*, *Leuconostoc*, *Enterococcus*, and *Streptococcus* spp.
Zimbabwe	Chikokivana	Maize and millet	Yeasts: *S. cerevisiae*
	Doro	Finger millet malt	Yeasts: *S. cerevisiae*, *Issatchenkia occidentalis*, *K. marxianus*, *C. glabrata*, *Sporobolomyces holsaticus*, and *Rhodotorula* spp.

**Table 2 foods-14-04240-t002:** Examples of microbial species included in commercial starter preparations used to produce fermented products.

Microbial Group	Application	Examples	References
LAB and staphylococci	Dairy Fermentation, Dairy-Based Beverages, and Kefir	*L. acidophilus*, *L. casei*, *L. plantarum*, *L. lactis*, *S.s cremoris*, *L. lactis* subsp. *lactis* biovar *diacetylact*, and *L. mesenteroides* subsp. *cremoris*	[152,153,154,155]
	Cheese Making	*L. lactis* subsp. *lactis*, *Lactococcus cremoris*, and *Streptococcus thermophilus*	[156,157,158]
	Dry Sausages and Cured Meats	*Latilactobacillus sakei*, *L. pentosus*, *L. casei*, *L. curvatus*, *L. plantarum*, *P.s acidilactici*, *P. pentosaceus*, *Kocuria varians* (Formerly known as *Micrococcus varians*), *Staphylococcus carnosus*, *Staphylococcus equorum*, and *Staphylococcus xylosus*	[159,160,161,162,163]
	Sauerkraut, Kimchi, and Pickles	*L. plantarum*, *L.mesenteroides*, *P. pentosaceus*, and *L. sakei*	[164,165]
	Plant-Based Beverages and Kombucha	*L. plantarum*, *Liquorilactobacillus nagelii*, and *Oenococcus oeni*	[166,167]
	Fermented Vegetable Juices	*L*. *plantarum*, *L. brevis*, and *L. sakei*	[168,169]
	Wine Making	*L. brevis*	[170,171]
Cutibacterium (formerly known as Propionibacterium)	Cheese Making	*Propionibacterium freudenreichii* and *Acidipropionibacterium acidipropionici*	[172,173,174]
Bifidobacteria	Various Dairy and Non-Dairy Products (health promoting role)	*Bifidobacterium animalis* and *Bifidobacterium bifidum*	[175]
Yeasts	Bread Making, pastry, and sourdough starters	*S. cerevisiae* (baker’s yeast)	[176]
	Brewing (Beer and Wine)	*S. cerevisiae*, *S. pastorianus*, and *S. uvarum*	[177]
	Cheese making	*C. famata* and *Yarrowia lipolytica*	[178]
	Fermented sausages	*C. famata* and *Y. lipolytica*	[179]
	Soy Sauce and Miso Beer (Sour/Farmhouse Ales)	*Brettanomyces bruxellensis*	[180]
Molds	Cheese Making	*Penicillium roqueforti* and *Penicillium camemberti* (Known as *Penicillium candidum*)	[181,182]
	Soy Sauce and Miso	*A. oryzae*	[183]

**Table 3 foods-14-04240-t003:** Features distinguishing natural from commercial starters.

Feature	Natural Starters	Commercial Starters
Microbial Diversity	High	Moderate to low
Consistency	Variable	High
Production Scale	Small-scale	From small- to large-scale
Production Environment	In the same place where they are used	Controlled facilities/laboratories
Certification	Not typically certified	Can be certified (e.g., organic)
Easiness of use	Depending on the starter	Yes
Most common application	Traditional/Artisanal foods	Industrial food production
Risk of pathogenic or spoilage microorganisms	From low to moderate	Tending to zero
Fermentation failure to due bacteriophages	Low risk	Moderate risk
Customization	Theoretically infinite (provided that operators with high education adapt the starter)	Relatively high

## Data Availability

No new data were created or analyzed in this study. Data sharing is not applicable to this article.

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
