# Peer review of "From Ancient Fermentations to Modern Biotechnology: Historical Evolution, Microbial Mechanisms, and the Role of Natural and Commercial Starter Cultures in Shaping Organic and Sustainable Food Systems"

_foods, 2025, doi:10.3390/foods14244240_

Round 1

Reviewer 1 Report

Comments and Suggestions for Authors

This review focuses on "fermentation starters for organic fermented foods and beverages," systematically tracing the historical origins and global application scenarios of natural starters (including grains, dairy, fruits and vegetables, seafood, and meats). It compares the core differences between natural and commercial starters, and, in conjunction with organic agriculture regulations and cutting-edge technologies (omics, synthetic biology), explores the potential applications and challenges of natural starters. The study aligns with the trends of "organic food industrialization" and "modernization of traditional fermentation technologies." The literature coverage is comprehensive (encompassing traditional fermented foods from multiple global regions, with a broad thematic scope and a strong integration of historical and contemporary perspectives), and the review is rich in case studies (Tables 1 and 2 include nearly one hundred fermented foods and their associated microorganisms). It effectively integrates regulatory and technological aspects, demonstrates forward-looking insights, and provides a systematic reference for the selection of fermentation starters and technological innovation in organic fermented foods. However, several issues remain that require further optimization to enhance the logical coherence and practical guidance of the review.

  1. Redundant and repetitive structure; need for improved logical hierarchy: In Section 2, "Different Categories of Fermented Foods," there is content overlap among subsections on cereal fermentation (sourdough, rice wine) and dairy fermentation (kefir, cheese)—for example, repeated discussion of the acidification function of lactic acid bacteria. Additionally, some paragraphs (such as 2.1.1, "History of Sourdough") duplicate content from Section 2, "Historical Origins." Moreover, in Section 7, "Challenges and Prospects," the subsection on "Technological Innovation" overlaps with Section 6, "Application of Natural Starters in Organic Processing," resulting in a loosely organized structure.

  2. Insufficient explanation of microbial mechanisms and disconnection between function and mechanism: The review frequently mentions "specific microorganisms involved in the fermentation of certain foods" (e.g., Bacillus species in fermented soy products) but does not explain "how microbial metabolic activities influence food quality." For instance, it merely states "lactic acid bacteria produce acid to extend shelf life" without elucidating the specific mechanism by which "lactic acid inhibits pathogens (such as Listeria) by lowering pH and disrupting cell membrane integrity." The "synergistic effects of microbial communities" in natural starters (e.g., mutualism between lactic acid bacteria and yeasts in sourdough) are described only at the phenomenon level, without explaining the metabolic interactions (e.g., yeast-produced ethanol serving as a carbon source for lactic acid bacteria, while lactic acid bacteria create a low-pH environment beneficial for yeasts).

  3. Non-standardized data presentation and figure/table labeling, leading to poor readability: In Table 1, the "microorganism" column shows inconsistent formatting of scientific names (e.g., "Limosilactobacillus fermentum" is sometimes labeled "formerly Lactobacillus fermentum," sometimes not), and for some products (such as the Indian "idli"), the microbial description is overly simplistic (mentioning only "lactic acid bacteria" without specifying species). Figures 1 and 2 (Italian organic starter products) show only product appearance without indicating "starter type (e.g., sourdough mother culture, kefir grains)" or core microbial composition, failing to reflect the relationship between products and their fermentation starters.

  4. Lack of industrial details for commercial starters and weak feasibility analysis: The review only mentions the advantages of commercial starters such as "standardization and high stability," without explaining "key industrial production processes" (e.g., choice of cryoprotectants, control of starter viability). Furthermore, it does not compare "cost differences between natural and commercial starters" (such as cost per unit of natural sourdough starter versus commercial freeze-dried starter), making it difficult to assess the industrial applicability of these technologies.

Overall, this review provides a comprehensive reference for the selection and technological innovation of fermentation starters in organic fermented foods, with abundant case studies and in-depth regulatory analysis. Optimizing structural redundancy, supplementing explanations of microbial mechanisms, and improving coverage of industrialization details would significantly enhance the scientific rigor and practical value of the review, offering clearer guidance for researchers and professionals in the food industry.

Author Response

  1. Redundant and repetitive structure; need for improved logical hierarchy: In Section 2, "Different Categories of Fermented Foods," there is content overlap among subsections on cereal fermentation (sourdough, rice wine) and dairy fermentation (kefir, cheese)—for example, repeated discussion of the acidification function of lactic acid bacteria. Additionally, some paragraphs (such as 2.1.1, "History of Sourdough") duplicate content from Section 2, "Historical Origins." Moreover, in Section 7, "Challenges and Prospects," the subsection on "Technological Innovation" overlaps with Section 6, "Application of Natural Starters in Organic Processing," resulting in a loosely organized structure.

CA: OK, thank you for this suggestion. Iterations were removed (ll. 869, 878) and we improved logical hierarchy, moving part of introduction in the paragraph 4 “The shift towards commercial starters” (ll. 632-637, 645-649, 658-663).

  1. Insufficient explanation of microbial mechanisms and disconnection between function and mechanism: The review frequently mentions "specific microorganisms involved in the fermentation of certain foods" (e.g., Bacillus species in fermented soy products) but does not explain "how microbial metabolic activities influence food quality." For instance, it merely states "lactic acid bacteria produce acid to extend shelf life" without elucidating the specific mechanism by which "lactic acid inhibits pathogens (such as Listeria) by lowering pH and disrupting cell membrane integrity." The "synergistic effects of microbial communities" in natural starters (e.g., mutualism between lactic acid bacteria and yeasts in sourdough) are described only at the phenomenon level, without explaining the metabolic interactions (e.g., yeast-produced ethanol serving as a carbon source for lactic acid bacteria, while lactic acid bacteria create a low-pH environment beneficial for yeasts).

CA: OK, thank you for this useful suggestion. Since microbial mechanisms that affect quality of fermented food items are very numerous, we have preferred to dedicate a new paragraph (“Microbial mechanisms within natural starters affecting food quality: sourdough as a paradigm”), but focusing just on those mechanisms that are typical of microorganisms dominating sourdough, a paradigmatic case of natural starter culture (ll.552-588).

  1. Non-standardized data presentation and figure/table labeling, leading to poor readability: In Table 1, the "microorganism" column shows inconsistent formatting of scientific names (e.g., "Limosilactobacillus fermentum" is sometimes labeled "formerly Lactobacillus fermentum," sometimes not), and for some products (such as the Indian "idli"), the microbial description is overly simplistic (mentioning only "lactic acid bacteria" without specifying species).

CA: OK, thank you. We have formatted names of microorganisms in Table 1; in addition, when details were available, we listed in Table 1 all the microbial species or genera.

Figures 1 and 2 (Italian organic starter products) show only product appearance without indicating "starter type (e.g., sourdough mother culture, kefir grains)" or core microbial composition, failing to reflect the relationship between products and their fermentation starters.

CA: OK, thank you. We have preferred to remove figure 1 and to modify figure 2, whose legend has been integrated with information about the starter type or, where available, microbial composition (ll. 658-663).

  1. Lack of industrial details for commercial starters and weak feasibility analysis: The review only mentions the advantages of commercial starters such as "standardization and high stability," without explaining "key industrial production processes" (e.g., choice of cryoprotectants, control of starter viability). Furthermore, it does not compare "cost differences between natural and commercial starters" (such as cost per unit of natural sourdough starter versus commercial freeze-dried starter), making it difficult to assess the industrial applicability of these technologies.

CA: Thank you for your comment. We unintentionally communicated that natural starters are cheaper than commercial ones. Probably in many cases, the opposite is true; for instance, because of low cost and ease of use, commercial baker’s yeast overcame traditional sourdough in the last century. Therefore, in the revised version, we have removed all the sentences that refer to cost of starters. Also based on a comment from the reviewer 2, specific for ll. 774-775 (old version), we profoundly revised the paragraph “Embracing nature: harnessing the potential of natural starters in organic food processing” (ll. 864-878).

Reviewer 2 Report

Comments and Suggestions for Authors

The manuscript “Starter cultures for producing organic fermented food and beverages” provides a comprehensive overview of natural microbial starters, focusing on their historical background and role in the fermentation of food and beverages. The topic addressed in this review is both timely and engaging, offering valuable insights into the role of natural starters in food fermentation, including organic food production. However, the manuscript requires several revisions to enhance its clarity, structure, and scientific rigor, as outlined in the comments below.

I recommend that the authors reconsider the current title of the manuscript, as it does not fully capture the breadth of the subject matter presented. In my view, the scope of the paper is considerably broader than the title suggests. A revised title would more accurately reflect the comprehensive range of topics addressed in the manuscript.

Lines 47-52: Please exercise greater caution when making broad statements about modern food preservation methods. While fermentation is a valuable and traditional method, many contemporary technologies effectively extend shelf life without relying on chemically synthesized additives.

Lines 110-112: Please consider revising the statement to reflect that fermentation is not suitable for preserving all types of food — for example, fats and oils are not appropriate substrates for fermentation due to their lack of fermentable sugars. Fermentation cannot replace refrigeration as a method of food preservation, as it alters the food’s composition, flavor, and texture, rather than maintaining its original state. Refrigeration is a universal technique that slows microbial growth without changing the product, making it suitable for a wide range of fresh and perishable foods. Each method serves a different purpose, and they are not interchangeable.

Chapter 2.2. The chapter on the application of fermentation in dairy processing is notably underdeveloped compared to, for example, Chapter 2.1. Fermentation has been widely used in dairy processing both historically and in modern times. Why did the authors choose to focus only on kefir, cheese, and dahi, omitting other significant fermented dairy products?

Chapter 2.2.2. In the description, the authors should clearly distinguish between acid-coagulated cheeses and rennet-coagulated cheeses. The production technologies for these two types of cheese are fundamentally different, and the role of microbial flora in curd formation varies significantly between them. 2.3. Fermented Horticultural Produce

Chapter 2.4 focuses on "Fermented beverages of vegetable and fruit origin." On what basis did the authors classify vinegar under this category (section 2.4.2)?

Lines 455-457: “Cider vinegar, in particular, is valued for its health-promoting properties and is believed to contribute to the maintenance of the body's pH balance when consumed regularly”. –  Can the authors provide scientific publications to support this statement? What do they mean by "regular consumption"? In the authors' opinion, are there any potential health risks associated with frequent vinegar intake?

2.6 To better reflect the chapter title, the text should concentrate more specifically on meat fermentation as a biological process, rather than combining it with other preservation methods such as drying, salting, or smoking.

Chapter 5: Could the authors provide a concise comparison table highlighting the similarities and differences between commercial and organic starter cultures?

Lines 760-763 – I do not quite understand the authors' intention — the same could be said about commercial cultures.

Lines 774-775: “natural starters can be produced using cheap and low impact protocols” - Could the authors comment on whether, given the challenges associated with producing natural starter cultures—such as ensuring product safety and consistent quality—their use in organic food production is truly economically viable? A brief analysis of cost-effectiveness in comparison to certified (organic) commercial cultures would be appreciated.

Conclusions: The conclusions should be more precise. They should clearly outline the benefits and risks associated with the use of natural starter cultures compared to certified commercial cultures used in organic agriculture. Additionally, the authors should explicitly indicate the current state of research regarding the feasibility of using natural cultures and identify which aspects still require further investigation.

Author Response

I recommend that the authors reconsider the current title of the manuscript, as it does not fully capture the breadth of the subject matter presented. In my view, the scope of the paper is considerably broader than the title suggests. A revised title would more accurately reflect the comprehensive range of topics addressed in the manuscript.

CA: OK, thank you; we have revised the title of the manuscript (ll.2-5).

Lines 47-52: Please exercise greater caution when making broad statements about modern food preservation methods. While fermentation is a valuable and traditional method, many contemporary technologies effectively extend shelf life without relying on chemically synthesized additives.

CA: OK, sorry for this oversimplification. We have removed the sentence about modern food preservation methods and modified the sentence about the general benefits of fermented food items (ll. 50-52).

Lines 110-112: Please consider revising the statement to reflect that fermentation is not suitable for preserving all types of food — for example, fats and oils are not appropriate substrates for fermentation due to their lack of fermentable sugars. Fermentation cannot replace refrigeration as a method of food preservation, as it alters the food’s composition, flavor, and texture, rather than maintaining its original state. Refrigeration is a universal technique that slows microbial growth without changing the product, making it suitable for a wide range of fresh and perishable foods. Each method serves a different purpose, and they are not interchangeable.

CA: OK, we have added a sentence that makes the limits of fermentation explicit (ll. 71-74).

Chapter 2.2. The chapter on the application of fermentation in dairy processing is notably underdeveloped compared to, for example, Chapter 2.1. Fermentation has been widely used in dairy processing both historically and in modern times. Why did the authors choose to focus only on kefir, cheese, and dahi, omitting other significant fermented dairy products?

CA: OK, we agree that this chapter was underdeveloped, compared to the one about fermented cereals and legumes. Our intention was not to provide an exhaustive overview of all fermented dairy products, but rather to highlight representative examples—kefir, cheese, and dahi—that illustrate the diversity of fermentation processes, microbial communities, and product characteristics within dairy fermentation. According to us, to extend the discussion, for instance, to the very large world of cheeses, would have diverted the reader’s attention from the aim of the review. However, we have developed that chapter a bit more, dedicating a sub-chapter to fermented dairy beverages, such as yogurt (ll. 332-350).

Chapter 2.2.2. In the description, the authors should clearly distinguish between acid-coagulated cheeses and rennet-coagulated cheeses. The production technologies for these two types of cheese are fundamentally different, and the role of microbial flora in curd formation varies significantly between them. 2.3. Fermented Horticultural Produce

CA: we are sorry, but we prefer to keep the current version, because the content of this chapter is merely historic.

Chapter 2.4 focuses on "Fermented beverages of vegetable and fruit origin." On what basis did the authors classify vinegar under this category (section 2.4.2)?

CA: OK, thank you for this question. We would like to clarify that although vinegar is used more as a condiment rather than as a beverage, it results from a two-stage fermentation not only of cereals but also of fruit- or vegetable-derived substrates. Therefore, we have chosen to discuss vinegar here, among other liquid fermented products of plant origin.

Lines 455-457: “Cider vinegar, in particular, is valued for its health-promoting properties and is believed to contribute to the maintenance of the body's pH balance when consumed regularly”. –  Can the authors provide scientific publications to support this statement? What do they mean by "regular consumption"? In the authors' opinion, are there any potential health risks associated with frequent vinegar intake?

CA: OK, thank you, we have modified the sentence and added a better explanation (ll. 455-457)

2.6 To better reflect the chapter title, the text should concentrate more specifically on meat fermentation as a biological process, rather than combining it with other preservation methods such as drying, salting, or smoking.

CA: OK, thank you. We have removed any direct hint to methods such as salting and drying (ll. 525-526, 544-545, 547).

Chapter 5: Could the authors provide a concise comparison table highlighting the similarities and differences between commercial and organic starter cultures?

CA: Sorry for having confused the reviewer. We did not want to compare commercial starter cultures with those commercial starter cultures usable for fermented organic food items, because the only difference consists in the fact that those usable for organic food items have to be cultured under conditions that meet organic certification standards (ll.708-709)

Lines 760-763 – I do not quite understand the authors' intention — the same could be said about commercial cultures.

CA: OK, thank you for this comment. We agree with you and therefore we have modified some sentences in this chapter (ll. 864-865, 868).

Lines 774-775: “natural starters can be produced using cheap and low impact protocols” - Could the authors comment on whether, given the challenges associated with producing natural starter cultures—such as ensuring product safety and consistent quality—their use in organic food production is truly economically viable? A brief analysis of cost-effectiveness in comparison to certified (organic) commercial cultures would be appreciated.

CA: Thank you for your comment, which agreed with one from the reviewer 1. We unintentionally communicated that natural starters are cheaper than commercial ones. Probably in many cases, the opposite is true; for instance, because of low cost and ease of use, commercial baker’s yeast overcame traditional sourdough in the last century. Therefore, in the revised version, we have removed all the sentences that refer to cost of starters. In addition, we have profoundly revised the paragraph “Embracing nature: harnessing the potential of natural starters in organic food processing” (ll. 864-878).

Conclusions: The conclusions should be more precise. They should clearly outline the benefits and risks associated with the use of natural starter cultures compared to certified commercial cultures used in organic agriculture. Additionally, the authors should explicitly indicate the current state of research regarding the feasibility of using natural cultures and identify which aspects still require further investigation.

CA: OK, we have written the conclusions, right from the start, following your suggestions, as well as those from the reviewer 3 (ll. 994-1019).

Reviewer 3 Report

Comments and Suggestions for Authors

The manuscript presents a well-written and comprehensive review of the use of starter cultures in the production of organic fermented foods and beverages. The topic is relevant and current, covering both scientific and regulatory aspects. 

It is recommended that the main objectives be clarified and a brief section describing the literature search strategy be included to improve transparency. Some sections could be condensed to avoid repetition, and the inclusion of summary tables or diagrams would facilitate understanding of the most important comparisons. It would also be beneficial to expand the discussion of microbial safety and regulatory requirements for organic certification, which would strengthen the article's impact.

It is recommended to clarify the relevance and/or novelty of this review compared to previous work.

The author guidelines mention following the PRISMA methodology for a literature review. Could you please include a brief description of the methodology used for the literature search and selection?

It is recommended to strengthen the conclusions section to highlight practical recommendations and future perspectives.

Author Response

It is recommended that the main objectives be clarified and a brief section describing the literature search strategy be included to improve transparency.

CA: OK. Thank you for this suggestion. We have clarified the main objectives of the review article (ll. 84-92) and dedicated a separate section for the literature strategy that has been used (ll. 93-104).

Some sections could be condensed to avoid repetition, and the inclusion of summary tables or diagrams would facilitate understanding of the most important comparisons.

CA: OK, thank you for this suggestion. We preferred to focus on one great difference between natural and commercial starters: consistency (in Table 3). To facilitate understanding of this difference, we have inserted a figure that exemplifies the concept with a case of a sourdough whose microbial community has undergone an important change (ll. 720-738). We think that other terms of comparison are easier to understand.

 It would also be beneficial to expand the discussion of microbial safety and regulatory requirements for organic certification, which would strengthen the article's impact.

CA: Ok. Thank you. We have added a detailed part discussing the microbial safety and regulatory requirements for organic certification (ll. 846-861).

It is recommended to clarify the relevance and/or novelty of this review compared to previous work.

CA: To our knowledge, previous reviews focused on one or few types of fermented food: De Vuyst et al 2023 dealt with sourdough leavened baked goods; Bassi et al 2015 dealt with cheeses and meat-based products. We have modified the text, considering those two papers, as well as a book chapter (ll. 75-83).

The author guidelines mention following the PRISMA methodology for a literature review. Could you please include a brief description of the methodology used for the literature search and selection?

CA: OK, thank you. Although we did not follow the PRISMA protocol, during the bibliographic search we placed emphasis on integrating recent findings and authoritative reviews to provide a comprehensive and coherent overview of the topic. We have modified the manuscript, adding a brief description of the methodology used for the literature search and selection (ll. 93-104).

It is recommended to strengthen the conclusions section to highlight practical recommendations and future perspectives.

CA: OK, we have written the conclusions, right from the start, following your suggestions, as well as those from the reviewer 2 (994-1019).

Round 2

Reviewer 1 Report

Comments and Suggestions for Authors

The author has revised the manuscript according to my suggestions, and I believe the changes are very satisfactory. At present, the manuscript can be accepted.